# CogVideoX: Text-to-Video Diffusion Models with An Expert Transformer

Zhuoyi Yang*‡    Jiayan Teng*‡    Wendi Zheng‡    Ming Ding†    Shiyu Huang†
Jiazheng Xu‡  Yuanming Yang‡  Wenyi Hong‡  Xiaohan Zhang†  Guanyu Feng†
Da Yin†      Yuxuan Zhang†     Weihan Wang†     Yean Cheng†     Bin Xu‡
Xiaotao Gu†     Yuxiao Dong‡     Jie Tang‡

†Zhipu AI    ‡Tsinghua University

**Text Prompt:** A lightning bolt shatters a mountaintop stone—out leaps the Monkey King in battle robes. Energy erupts, winds howl.

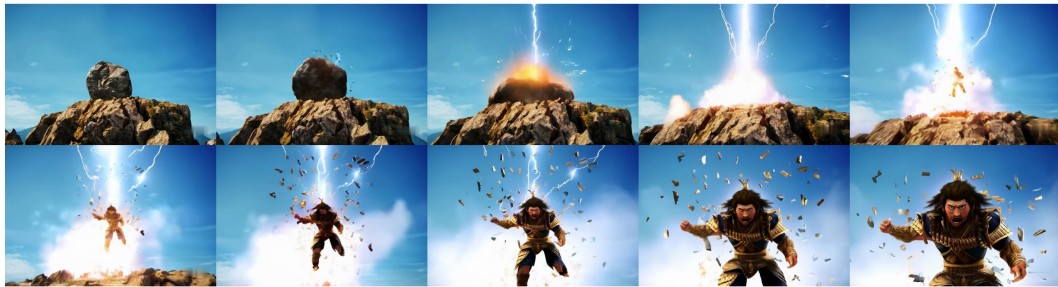

**Text Prompt:** A bald man put on a colorful wig.

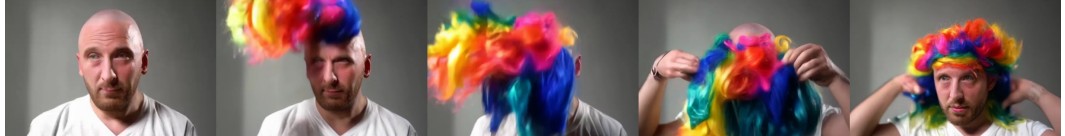

Figure 1: CogVideoX can generate long-duration, high-resolution videos with coherent actions and rich semantics.

## Abstract

We present CogVideoX, a large-scale text-to-video generation model based on diffusion transformer, which can generate 10-second continuous videos that align seamlessly with text prompts, with a frame rate of 16 fps and resolution of $768\times 1360$ pixels. Previous video generation models often struggled with limited motion and short durations. It is especially difficult to generate videos with coherent narratives based on text. We propose several designs to address these issues. First, we introduce a 3D Variational Autoencoder (VAE) to compress videos across spatial and temporal dimensions, enhancing both the compression rate and video fidelity. Second, to improve text-video alignment, we propose an expert transformer with expert adaptive LayerNorm to facilitate the deep fusion between the two modalities. Third, by employing progressive training and multi-resolution frame packing, CogVideoX excels at generating coherent, long-duration videos with diverse shapes and dynamic movements. In addition, we develop an effective pipeline that includes various pre-processing strategies for text and video data. Our innovative video captioning model significantly improves generation quality and semantic alignment. Results show that CogVideoX achieves state-of-the-art performance in both automated benchmarks and human evaluation. We publish the code and model checkpoints

---

*Equal contributions. Core contributors: Zhuoyi, Jiayan, Wendi, Ming, Shiyu and Xiaotao.
{yangzy22,tengjy24}@mails.tsinghua.edu.cn, corresponding author: jietang@tsinghua.edu.cn
Visiting our demo website https://yzy-thu.github.io/CogVideoX-demo/ to watch more videos!

of CogVideoX along with our VAE model and video captioning model at https://github.com/THUDM/CogVideo.

# 1 INTRODUCTION

The rapid development of text-to-video models has been phenomenal, driven by both the Transformer architecture (Vaswani et al., 2017) and diffusion model (Ho et al., 2020). Early attempts to pretrain and scale Transformers to generate videos from text have shown great promise, such as CogVideo (Hong et al., 2022) and Phenaki (Villegas et al., 2022). Meanwhile, diffusion models have recently made exciting advancements in video generation(Singer et al., 2022; Ho et al., 2022). By using Transformers as the backbone of diffusion models, i.e., Diffusion Transformers (DiT) (Peebles & Xie, 2023), text-to-video generation has reached a new milestone, as evidenced by the impressive Sora showcases (OpenAI, 2024).

Despite these rapid advancements in DiTs, it remains technically unclear how to achieve long-term consistent video generation with dynamic plots. For example, previous models had difficulty generating a video based on a prompt like "a bolt of lightning splits a rock, and a person jumps out from inside the rock".

In this work, we train and introduce CogVideoX, a set of large-scale diffusion transformer models designed for generating long-term, temporally consistent videos with rich motion semantics. We address the challenges mentioned above by developing a 3D Variational Autoencoder, an expert Transformer, a progressive training pipeline, and a video data filtering and captioning pipeline, respectively.

First, to efficiently consume high-dimension video data, we design and train a 3D causal VAE that compresses the video along both spatial and temporal dimensions. Compared to previous method(Blattmann et al., 2023) of fine-tuning 2D VAE, this strategy helps significantly reduce the sequence length and associated training compute and also helps prevent flicker in the generated videos, that is, ensuring continuity among frames.

Second, to improve the alignment between videos and texts, we propose an expert Transformer with expert adaptive Layer-Norm to facilitate the fusion between the two modalities. To ensure the temporal consistency in video generation and capture large-scale motions, we propose to use 3D full attention to comprehensively model the video along both temporal and spatial dimensions.

Third, as most video data available online lacks accurate textual descriptions, we develop a video captioning pipeline capable of accurately describing video content. This pipeline is used to generate new textual descriptions for all video training data, which significantly enhances CogVideoX's ability to grasp precise semantic understanding.

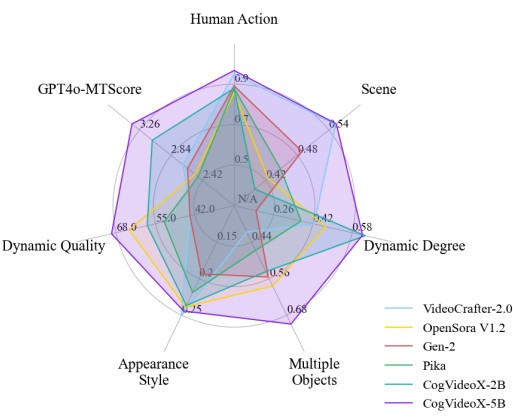

Figure 2: The performance of openly-accessible text-to-video models in different aspects.

In addition, we adopt and design progressive training techniques, including multi-resolution frame pack and resolution progressive training, to further enhance the generation performance and stability of CogVideoX. Furthermore, we propose Explicit Uniform Sampling, which stablizes the training loss curve and accelerates convergence by setting different timestep sampling intervals on each data parallel rank.

To date, we have completed the CogVideoX training with two sizes: 5 billion and 2 billion, respectively. Both machine and human evaluations suggest that CogVideoX-5B outperforms well-known video models and CogVideoX-2B is very competitive across most dimensions.

Figure 2 shows the performance of CogVideoX-5B and CogVideoX-2B in different aspects. It shows that CogVideoX has the property of being scalable. As the size of model parameters, data volume, and training volume increase, the performance will get better in the future.

Our contributions can be summarized as follows:

- We propose CogVideoX, a simple and scalable structure with a 3D causal VAE and an expert transformer, designed for generating coherent, long-duration, high-action videos. It can generate long videos with multiple aspect ratios, up to 768×1360 resolution, 10 seconds in length, at 16fps.

- We evaluate CogVideoX through automated metric evaluation and human assessment, compared with openly-accessible top-performing text-to-video models. CogVideoX achieves state-of-the-art performance.

- We publicly release our 5B and 2B models, including text-to-video and image-to-video versions, the first commercial-grade open-source video generation models. We hope it can advance the filed of video generation.

## 2 THE COGVIDEOX ARCHITECTURE

In the section, we present the CogVideoX model. Figure 3 illustrates the overall architecture. Given a pair of video and text input, we design a **3D causal VAE** to compress the video into the latent space, and the latents are then patchified and unfolded into a long sequence denoted as $z_{\text{vision}}$. Simultaneously, we encode the textual input into text embeddings $z_{\text{text}}$ using T5 (Raffel et al., 2020). Subsequently, $z_{\text{text}}$ and $z_{\text{vision}}$ are concatenated along the sequence dimension. The concatenated embeddings are then fed into a stack of **expert transformer** blocks. Finally, the model output are unpatchified to restore the original latent shape, which is then decoded using a 3D causal VAE decoder to reconstruct the video. We illustrate the technical design of the 3D causal VAE and expert transfomer in detail.

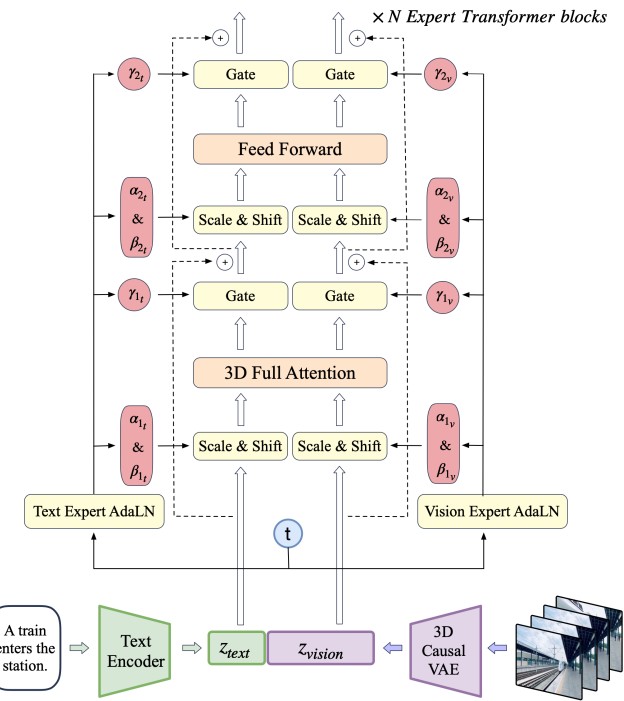

Figure 3: **The overall architecture of CogVideoX.**

## 2.1 3D CAUSAL VAE

Videos contain both spatial and temporal information, typically resulting in much larger data volumes than images. To tackle the computational challenge of modeling video data,

Table 1: Ablation with different variants of 3D VAE. The baseline is SDXL(Podell et al., 2023) 2D VAE. Flickering calculates the L1 difference between each pair of adjacent frames to evaluate the degree of flickering in the video. We use variant B for pretraining.

| Variants | Baseline | A | B | C | D | E |
|---|---|---|---|---|---|---|
| Compression | 8×8×1 | 8×8×4 | 8×8×4 | 8×8×4 | 8×8×8 | 16×16×8 |
| Latent channel | 4 | 8 | 16 | 32 | 32 | 128 |
| Flickering↓ | 93.2 | 87.6 | 86.3 | 87.7 | 87.8 | 87.3 |
| PSNR↑ | 28.4 | 27.2 | 28.7 | 30.5 | 29 | 27.9 |

we propose to implement a video compression module based on 3D Variational Autoencoders (Yu et al., 2023b). The idea is to incorporate three-dimentional convolutions to compress videos both spatially and temporally. This can help achieve a higher compression ratio with largely improved quality and continuity of video reconstruction.

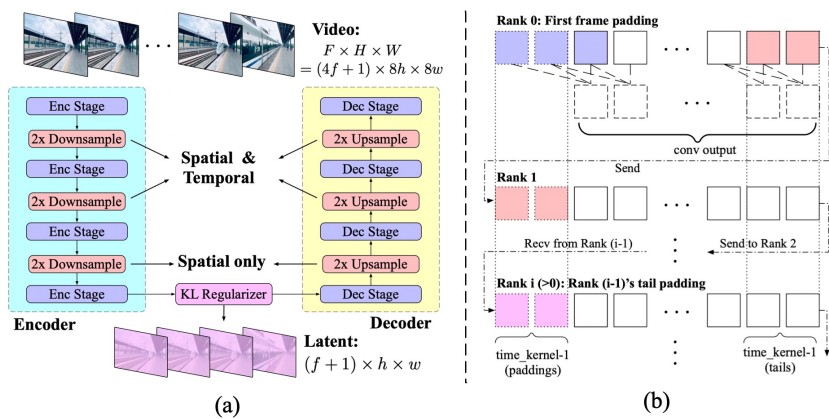

Figure 4: (a) The structure of the 3D VAE in CogVideoX. It comprises an encoder, a decoder and a latent space regularizer, achieving a 8×8×4 compression from pixels to the latents. (b) The context parallel implementation on the temporally causal convolution.

Figure 4 (a) shows the structure of the proposed 3D VAE. It comprises an encoder, a decoder and a Kullback-Leibler (KL) regularizer. The encoder and decoder consist of symmetrically arranged stages, respectively performing $2\times$ downsampling and upsampling by the interleaving of ResNet block stacked stages. Some blocks perform 3D downsampling (upsampling), while others only perform 2D downsampling (upsampling).

We adopt the temporally causal convolution (Yu et al., 2023b), which places all the paddings at the beginning of the convolution space, as shown in Figure 4 (b). This ensures that future information does not influence the present or past predictions.

We also conducted ablation studies comparing different compression ratios and latent channels in table 1. After using 3D structures, the reconstructed video shows almost no more jitter, and as the latent channels increase, the restoration quality improves. However, when spatial-temporal compression is too aggressive (16×16×8), even if the channel dimensions are correspondingly increased, the convergence of the model also becomes extremely difficult. Exploring VAE with larger compression ratios is our future work.

Given that processing long-duration videos introduces excessive GPU memory usage, we apply context parallel at the temporal dimension for 3D convolution to distribute computation among multiple devices. As illustrated by Figure 4 (b), due to the causal nature of the convolution, each rank simply sends a segment of length $k - 1$ to the next rank, where $k$ indicates the temporal kernel size. This results in relatively low communication overhead.

During training, we first train a 3D VAE at $256 \times 256$ resolution and 17 frames to save computation. Randomly select 8 or 16 fps to enhance the model's robustness. We observe

that the model can encode larger resolution videos well without additional training as it has no attention modules, but this isn't effective when encoding videos with more frames.

Therefore, we conduct a two-stage training by first training on 17-frame videos and finetuning by context parallel on 161-frame videos. Both stages utilize a weighted combination of the L1 reconstruction loss, LPIPS (Zhang et al., 2018) perceptual loss, and KL loss. After a few thousand training steps, we additionally introduce the GAN loss from a 3D discriminator.

## 2.2 EXPERT TRANSFORMER

We introduce the design choices in Transformer for CogVideoX, including the patching, positional embedding, and attention strategies.

**Patchify.** The 3D causal VAE encodes a video latent of shape $T \times H \times W \times C$, where $T$ represents the number of frames, $H$ and $W$ represent the height and width of each frame, $C$ represents the channel number, respectively. The video latents are then patchified, generating sequence $z_{\text{vision}}$ of length $\frac{T}{q} \cdot \frac{H}{p} \cdot \frac{W}{p}$. When $q > 1$, we repeat the first frame of videos and images at the beginning of the sequence to enable joint training of images and videos.

**3D-RoPE.** Rotary Position Embedding (RoPE) (Su et al., 2024) is a relative positional encoding that has been demonstrated to capture inter-token relationships effectively in LLMs, particularly excelling in modeling long sequences. To adapt to video data, we extend the original RoPE to 3D-RoPE. Each latent in the video tensor can be represented by a 3D coordinate $(x, y, t)$. We independently apply 1D-RoPE to each dimension of the coordinates, each occupying 3/8, 3/8, and 2/8 of the hidden states' channel. The resulting encoding is then concatenated along the channel dimension to obtain the final 3D-RoPE encoding.

**Expert Adaptive Layernorm.** We concatenate the embeddings of both text and video at the input stage to better align visual and semantic information. However, the feature spaces of these two modalities differ significantly, and their embeddings may even have different numerical scales. To better process them within the same sequence, we employ the Expert Adaptive Layernorm to handle each modality independently. As shown in Figure 3, following DiT (Peebles & Xie, 2023), we use the timestep $t$ of the diffusion process as the input to the modulation module. Then, the Vision Expert Adaptive Layernorm (Vison Expert AdaLN) and Text Expert Adaptive Layernorm (Text Expert AdaLN) apply this modulation to the vision hidden states and text hidden states, respectively. This strategy promotes the alignment of feature spaces across two modalities while minimizing additional parameters.

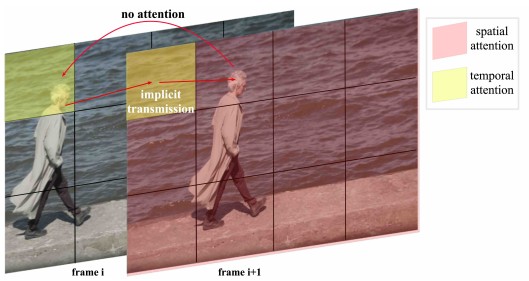

Figure 5: The separated spatial and temporal attention makes it challenging to handle the large motion between adjacent frames. In the figure, the head of the person in frame $i + 1$ cannot directly attend to the head in frame $i$. Instead, visual information can only be implicitly transmitted through other background patches. This can lead to inconsistency issues in the generated videos.

**3D Full Attention.** Previous works (Singer et al., 2022; Guo et al., 2023) often employ separated spatial and temporal attention to reduce computational complexity and facilitate fine-tuning from text-to-image models. However, as illustrated in Figure 5, this separated attention approach requires extensive implicit transmission of visual information, significantly increasing the learning complexity and making it challenging to maintain the consistency of large-movement objects. Considering the great success of long-context training in LLMs (AI@Meta, 2024) and the efficiency of FlashAttention (Dao et al., 2022), we propose a 3D text-video hybrid attention mechanism. This mechanism not only achieves better results but can also be easily adapted to various parallel acceleration methods.

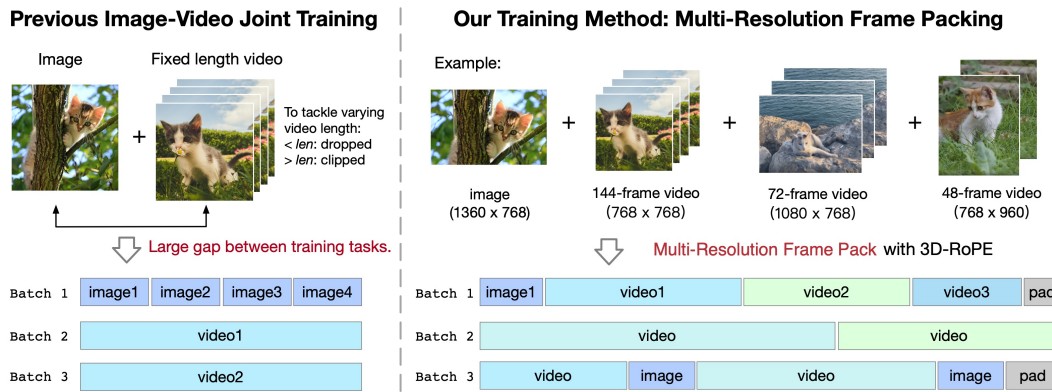

Figure 6: The diagram of mixed-duration training and Frame Pack. To fully utilize the data and enhance the model's generalization capability, we train on videos of different duration within the same batch.

## 3 Training CogVideoX

We mix images and videos during training, treating each image as a single-frame video. Additionally, we employ progressive training from the resolution perspective. For the diffusion setting, we adopt v-prediction (Salimans & Ho, 2022) and zero SNR (Lin et al., 2024), following the noise schedule used in LDM (Rombach et al., 2022).

### 3.1 Multi-Resolution Frame Pack

Previous video training methods often involve joint training of images and videos with a fixed number of frames (Singer et al., 2022; Blattmann et al., 2023). However, this approach usually leads to two issues: First, there is a significant gap between the two input types using bidirectional attention, with images having one frame while videos having dozens of frames. We observe that models trained this way tend to diverge into two generative modes based on the token count and not to have good generalizations. Second, to train with a fixed duration, we have to discard short videos and truncate long videos, which prevents full utilization of the videos of varying number of frames. For different resolutions, SDXL(Podell et al., 2023) uses a bucketed approach to address the issue of generating cropped images, but it makes the data and training pipeline more complex.

To address these issues, we chose mixed-duration training, which means training videos of different lengths together. However, inconsistent data shapes within the batch make training difficult. Inspired by Patch'n Pack (Dehghani et al., 2024), we place videos of different duration (also different resolutions) into the same batch to ensure consistent shapes within each batch, a method we refer to as *Multi-Resolution Frame Pack*, illustrated in Figure 6.

We use 3D RoPE to model the position relationship of various video shape. There are two ways to adapt RoPE to different resolutions and durations. One approach is to expand the position encoding table and, for each video, select the front portion of the table according to the resolution (extrapolation). The other is to scale a fixed-length position encoding table to match the resolution of the video (interpolation). Considering that RoPE is a relative position encoding, we chose the first approach to keep the clarity of model details.

### 3.2 Progressive Training

Videos from the Internet usually include a significant amount of low-resolution ones. And directly training on high-resolution videos is extremely expensive. To fully utilize data and save costs, the model is first trained on 256px videos to learn semantic and low-frequency knowledge. Then it is trained on gradually increased resolutions, from 256px to 512px, 768px, to learn high-frequency knowledge. To maintain the ability of generating videos with different aspect ratios, we keep the aspect ratio unchanged and resize the short side to above

resolutions. Finally, we do a high-quality fine-tuning, See Appendix A Moreover, we trained an image-to-video model based on above model. See Appendix D for details.

## 3.3 EXPLICIT UNIFORM SAMPLING

Ho et al. (2020) defines the training objective of diffusion as

$$L_{\text{simple}}(\theta) := \mathbf{E}_{t,x_0,\epsilon} \left\| \epsilon - \epsilon_\theta(\sqrt{\bar{\alpha}_t}x_0 + \sqrt{1-\bar{\alpha}_t}\epsilon, t) \right\|^2, \tag{1}$$

where $t$ is uniformly distributed between 1 and T. The common practice is for each rank in the data parallel group to uniformly sample a value between 1 and $T$, which is in theory equivalent to Equation 1. However, in practice, the results obtained from such random sampling are often not sufficiently uniform, and since the magnitude of the diffusion loss is related to the timesteps, this can lead to significant fluctuations in the loss. Thus, we propose to use *Explicit Uniform Sampling* to divide the range from 1 to $T$ into $n$ intervals, where $n$ is the number of ranks. Each rank then uniformly samples within its respective interval. This method ensures a more uniform distribution of timesteps. As shown in Figure 10 (d), the loss curve from training with Explicit Uniform Sampling is noticeably more stable.

## 3.4 DATA

We construct a collection of relatively high-quality video clips with text descriptions with video filters and recaption models. After filtering, approximately 35M single-shot clips remain, with each clip averaging about 6 seconds. We additionally use 2B images filtered with aesthetics score from LAION-5B (Schuhmann et al., 2022) and COYO-700M (Byeon et al., 2022) datasets to assist training.

**Video Filtering.** Video generation models should capture the dynamic nature of the world. However, raw video data often contains significant noise for two intrinsic reasons: First, the artificial editing during video creation can distort the true dynamic information; Second, video quality may suffer due to filming issues such as camera shakes or using subpar equipment. In addition to the intrinsic quality of the videos, we also consider how well the video data supports model training. Videos with minimal dynamic information or lacking connectivity in dynamic aspects are considered detrimental. Consequently, we have developed a set of negative labels, which include:

- **Editing**: Videos that have undergone noticeable artificial processing, such as re-editing and special effects, which compromise the visual integrity.
- **Lack of Motion Connectivity**: Video segments with transitions that lack coherent motion, often found in artificially spliced videos or those edited from static images.
- **Low Quality**: Poorly shot videos with unclear visuals or excessive camera shake.
- **Lecture Type**: Videos focusing primarily on a person continuously talking with minimal effective motion, such as lectures, and live-streamed discussions.
- **Text Dominated**: Videos containing a large amount of visible text or primarily focusing on textual content.
- **Noisy Screenshots**: Videos captured directly from phone or computer screens, often characterized by poor quality.

We first sample 20,000 videos and label each video as positive or negative by their quality. Using these annotations, we train 6 filters based on Video-LLaMA (Zhang et al., 2023b) to screen out low-quality video data. Examples of negative labels and the classifier's performance on the test set can be found in appendix K. In addition, we calculate the optical flow scores and image aesthetic scores of all training videos, and dynamically adjust their threshold during training to ensure the dynamic and aesthetic quality of generated videos.

**Video Captioning.** Video-text pairs are essential for the training of text-to-video generation models. However, most video data does not come with corresponding descriptive text.

Therefore, it is necessary to label the video data with comprehensive textual descriptions. There are some video caption datasets available now, such as Panda70M (Chen et al., 2024b), COCO Caption (Lin et al., 2014), and WebVid Bain et al. (2021b). However, the captions in these datasets are usually very short and fail to describe the video comprehensively.

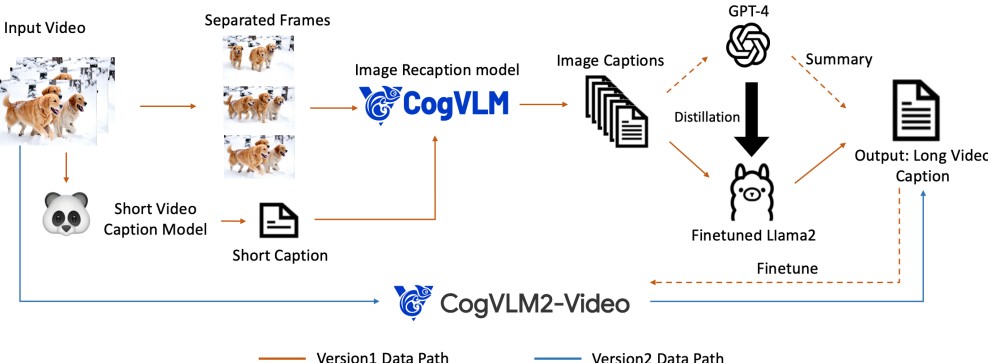

Figure 7: The pipeline for dense video caption data generation. In this pipeline, we generate short video captions with the Panda70M model, extract frames to create dense image captions, and use GPT-4 to summarize these into final video captions. To accelerate this process, we fine-tuned a Llama 2 model with the GPT-4 summaries.

To generate high-quality video caption data, we establish a *Dense Video Caption Data Generation* pipeline, as detailed in Figure 7. The main idea is to generate video captions with the help of image captions.

First, we use the video caption model from Chen et al. (2024b) to generate short captions for the videos. Then, we employ the image recaptioning model CogVLM (Wang et al., 2023a) used in CogView3 (Zheng et al., 2024a) to create dense image captions for each frame. Subsequently, we use GPT-4 to summarize all the image captions to produce the final video caption. To accelerate the generation from image captions to video captions, we fine-tune a LLaMA2 (Touvron et al., 2023) using the summary data generated by GPT-4 (Achiam et al., 2023), enabling large-scale video caption data generation. Additional details regarding the video caption data generation process can be found in Appendix G.

To further accelerate video recaptioning, we also fine-tune an end-to-end video understanding model CogVLM2-Caption , based on the CogVLM2-Video (Hong et al., 2024) and Llama3 (AI@Meta, 2024), by using the dense caption data generated from the aforementioned pipeline. Examples of video captions generated by this end-to-end CogVLM2-Caption model are shown in fig. 15 and Appendix H. CogVLM2-Caption can provide detailed descriptions of video content and changes. Interestingly, we find that we can perform video-to-video generation by connecting CogVideoX and CogVLM2-Caption, as detailed in appendix I.

## 4 EXPERIMENTS

### 4.1 ABLATION STUDY

We conducted ablation studies on some of the designs mentioned in Section 2 to verify their effectiveness.

**Position Embedding.** We compared 3D RoPE with sinusoidal absolute position embedding. As shown in Figure 10a indicates the loss curve of RoPE converges significantly faster than absolute one. This is consistent with the common choice in LLMs.

**Expert Adaptive Layernorm.** We compare three architectures in Figure 8a, 8d and Figure 10c: MMDiT Esser et al. (2024), Expert AdaLN(CogVideoX), without Expert AdaLN.

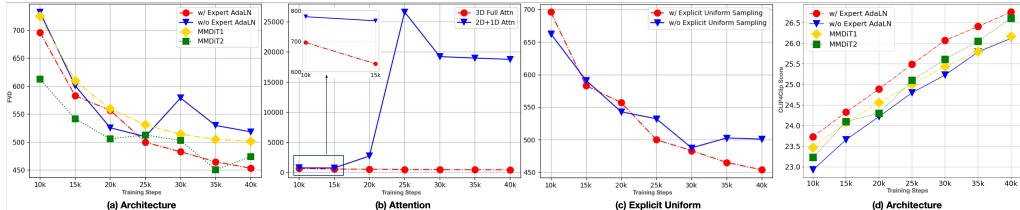

Figure 8: Ablation studies on WebVid test dataset with 500 videos. MMDiT1 has the same number of parameters with the expert AdaLN. MMDiT2 has the same number of layers but twice number of parameters. a, b, c measure FVD, d measures CLIP4Clip score.

Cross-attention DiT has been shown to be inferior to MMDiT in (Esser et al., 2024), so we don't repeat. According to FVD, CLIP4Clip(Luo et al., 2022) Score and loss, expert AdaLN significantly outperforms the model without expert AdaLN and MMDiT with the same number of parameters. We infer that expert adaptive layernorm is enough to alleviate the difference in feature space between the two modalities. So two independent transformers in MMDiT are not necessary, which greatly increases the number of parameters. Moreover, the design of Expert AdaLN is more simplified than MMDiT and is closer to current LLMs, making it easier to scale up further.

**3D Full Attention.**    In Figure 8b and Figure 10b, when we replace 3D full attention with 2D + 1D attention, the FVD will become much higher than 3D attention in early steps. We also observe that 2D+1D is unstable and prone to collapse. We suppose that as the model size increases, such as 5B, training becomes more prone to instability, placing higher demands on the structural design. The 2D+1D structure, as discussed in section 2.2, is not suitable for video generation tasks, which could lead to instability during training.

**Explicit Uniform Sampling.**    From Figure 8c and Figure 10d, we find that using Explicit Uniform Sampling can make a more stable decrease in loss and get a better performance. In addition, in Table 9 we compare the loss at each diffusion timestep alone between two choices for a more precise comparison. We find that the loss at all timesteps is lower with explicit uniform sampling, indicating that this method can also accelerate loss convergence. We suppose that this is because the loss of different timesteps varies greatly. When the timesteps sampled for training are not uniform enough, the loss fluctuates greatly due to the above randomness. Explicit uniformity can reduce randomness, thereby bringing a common decrease in all timesteps.

## 4.2   EVALUATION

### 4.2.1   AUTOMATED METRIC EVALUATION

**VAE Reconstruction Effect**    We compared our 3DVAE with other open-source 3DVAE on 256 × 256 resolution 17-frame videos, using the validation set of the WebVid (Bain et al., 2021a). On table 2, our VAE achieved the best PSNR and exhibited the least jitter. Notably, other VAE methods use fewer latent channels than ours.

**Evaluation Metrics.**    To evaluate the text-to-video generation, we employ several metrics in Vbench (Huang et al., 2024) that are consistent with human perception: *Human Action*, *Scene*, *Dynamic Degree*, *Multiple Objects*, and *Appearance Style*. Other metrics, such as color, tend to give higher scores to simple, static videos, so we do not use them.

Table 2: Comparison with the performance of other spatiotemporal compression VAEs.

|  | Flickering ↓ | PSNR ↑ |
|---|---|---|
| Open-Sora | 92.4 | 28.5 |
| Open-Sora-Plan | 90.2 | 27.6 |
| Ours | **85.5** | **29.1** |

Table 3: Evaluation results of CogVideoX-5B and CogVideoX-2B.

| Models | Human Action | Scene | Dynamic Degree | Multiple Objects | Appear. Style | Dynamic Quality | GPT4o-MT Score |
|---|---|---|---|---|---|---|---|
| T2V-Turbo(Li et al., 2024) | 95.2 | **55.58** | 49.17 | 54.65 | 24.42 | – | – |
| AnimateDiffGuo et al. (2023) | 92.6 | 50.19 | 40.83 | 36.88 | 22.42 | – | 2.62 |
| VideoCrafter-2.0(Chen et al., 2024a) | 95.0 | 55.29 | 42.50 | 40.66 | **25.13** | 43.6 | 2.68 |
| OpenSora V1.2(Zheng et al., 2024b) | 85.8 | 42.47 | 47.22 | 58.41 | 23.89 | 63.7 | 2.52 |
| Show-1(Zhang et al., 2023a) | 95.6 | 47.03 | 44.44 | 45.47 | 23.06 | 57.7 | – |
| Gen-2(runway, 2023) | 89.2 | 48.91 | 18.89 | 55.47 | 19.34 | 43.6 | 2.62 |
| Pika(pik, 2023) | 88.0 | 44.80 | 37.22 | 46.69 | 21.89 | 52.1 | 2.48 |
| LaVie-2(Wang et al., 2023b) | 96.4 | 49.59 | 31.11 | 64.88 | 25.09 | – | 2.46 |
| **CogVideoX-2B** | 96.6 | 55.35 | **66.39** | 57.68 | 24.37 | 57.7 | 3.09 |
| **CogVideoX-5B** | **96.8** | 55.44 | 62.22 | **70.95** | 24.44 | **69.5** | **3.36** |

For longer-generated videos, some models might produce videos with minimal changes between frames to get higher scores, but these videos lack rich content. Therefore, metrics for evaluating the dynamism become important. To address this, we use two video evaluation tools: *Dynamic Quality* (Liao et al., 2024) and *GPT4o-MTScore* (Yuan et al., 2024).

*Dynamic Quality* is defined by the integration of various quality metrics with dynamic scores, mitigating biases arising from negative correlations between video dynamics and video quality. GPT4o-MTScore is a metric designed to measure the metamorphic amplitude of time-lapse videos using GPT-4o, such as those depicting physical, biological, and meteorological changes.

**Results.** Table 3 provides the performance comparison of CogVideoX and other models. CogVideoX-5B achieves the best performance in five out of the seven metrics and shows competitive results in the remaining two metrics. These results demonstrate that the model not only excels in video generation quality but also outperforms previous models in handling various complex dynamic scenes. In addition, Figure 2 presents a radar chart that visually illustrates the performance advantages of CogVideoX. We present the time and space consumption during inference at different resolutions in appendix A.

### 4.2.2 HUMAN EVALUATION

In addition to automated scoring mechanisms, we also establish a comprehensive human evaluation framework to assess the general capabilities of video generation models. Evaluators will score the generated videos on four aspects: Sensory Quality, Instruction Following, Physics Simulation, and Cover Quality, using three levels: 0, 0.5, or 1. Each level is defined by detailed guidelines. The specific details are provided in the Appendix J.

We compare Kling (2024.7), one of the best closed-source models, with CogVideoX-5B under this framework. The results shown in Table 4 indicate that CogVideoX-5B wins the human preference over Kling across all aspects.

Table 4: Human evaluation between CogVideoX and Kling.

| Model | Sensory Quality | Instruction Following | Physics Simulation | Cover Quality | Total Score |
|---|---|---|---|---|---|
| Kling | 0.638 | 0.367 | 0.561 | 0.668 | 2.17 |
| **CogVideoX-5B** | **0.722** | **0.495** | **0.667** | **0.712** | **2.74** |

## 5 CONCLUSION

In this paper, we present CogVideoX, a state-of-the-art text-to-video diffusion model. It leverages a 3D VAE and an Expert Transformer architecture to generate coherent long duration videos with significant motion. We are also exploring the scaling laws of video generation models and aim to train larger and more powerful models to generate longer and higher-quality videos, pushing the boundaries of what is achievable in text-to-video generation.

ACKNOWLEDGMENTS

This work is supported by NSFC 62425601 and 62495063, Tsinghua University Initiative Scientific Research Program 20233080067, New Cornerstone Science Foundation through the XPLORER PRIZE. We would like to thank all the data annotators, infrastructure operators, collaborators, and partners. We also extend our gratitude to everyone at Zhipu AI and Tsinghua University who have provided support, feedback, or contributed to the CogVideoX, even if not explicitly mentioned in this report. We would also like to greatly thank BiliBili for technical discussions.

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

Appendix Contents

## A  Training Details

**High-Quality Fine-Tuning.**   Since the filtered pre-training data still contains a certain proportion of dirty data, such as subtitles, watermarks, and low-bitrate videos, we selected a subset of higher quality video data, accounting for 20% of the total dataset, for fine-tuning in the final stage. This step effectively removed generated subtitles and watermarks and slightly improved the visual quality. However, we also observed a slight degradation in the model's semantic ability.

**Visualizing different rope interpolation methods**   When adapting low-resolution position encoding to high-resolution, we consider two different methods: interpolation and extrapolation. We show the effects of two methods in Figure 9. Interpolation tends to preserve global information more effectively, whereas the extrapolation better retains local details. Given that RoPE is a relative position encoding, We chose the extrapolation to maintain the relative position between pixels.

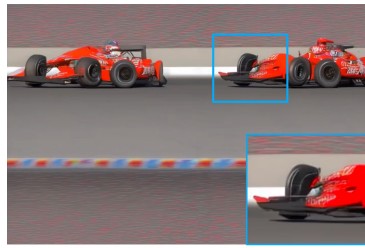 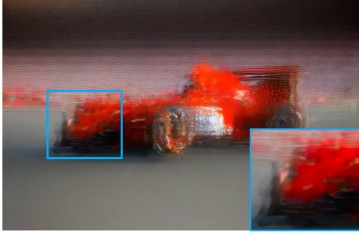

RoPE Extrapolation                RoPE Interpolation

Figure 9: The comparison between the initial generation states of extrapolation and interpolation when increasing the resolution with RoPE. Extrapolation tends to generate multiple small, clear, and repetitive images, while interpolation generates a blurry large image.

**Model & Training Hyperparameters**   We present the model and training hyperparameters in table 5 and table 6.

| Training Stage | stage1 | stage2 | stage3 | stage4 (FT) |
|---|---|---|---|---|
| Max Resolution | 256×384 | 480×720 | 768×1360 | 768×1360 |
| Max duration | 6s | 6s | 10s | 10s |
| Batch Size | 2000 | 1000 | 250 | 100 |
| Sequence Length | 25k | 75k | 700k | 700k |
| Training Steps | 400k | 220k | 120k | 10k |

Table 5: Hyperparameters of CogvideoX-2b and CogVideo-5b.

| Hyperparameter | CogvideoX-2b | CogVideo-5b |
|---|---|---|
| Number of Layers | 30 | 42 |
| Attention heads | 32 | 48 |
| Hidden Size | 1920 | 3072 |
| Position Encoding | sinusoidal | RoPE |
| Time Embedding Size | 256 | |
| Weight Decay | 1e-4 | |
| Adam $\epsilon$ | 1e-8 | |
| Adam $\beta_1$ | 0.9 | |
| Adam $\beta_2$ | 0.95 | |
| Learning Rate Decay | cosine | |
| Gradient Clipping | 1.0 | |
| Text Length | 226 | |
| Max Sequence Length | 82k | |
| Lowest aesthetic-value | 4.5 | |
| Training Precision | BF16 | |

Table 6: Hyperparameters of CogvideoX-2b and CogVideo-5b.

| | 5b-480x720-6s | 5b-768x1360-5s | 2b-480x720-6s | 2b-768x1360-5s |
|---|---|---|---|---|
| Time | 113s | 500s | 49s | 220s |
| Memory | 26GB | 76GB | 18GB | 53GB |

Table 7: Inference time and memory consumption of CogVideoX. We evaluate the model on bf, H800 with 50 inference steps.

| | 256*384*6s | 480*720*6s | 768*1360*5s |
|---|---|---|---|
| 2D+1D | 0.38s | 1.26s | 4.17s |
| 3D | 0.41s | 2.11s | 9.60s |

Table 8: Inference time comparison between 3D Full attention and 2D+1D attention. We evaluate the model on bf, H800 with one dit forward step. Thanks to the optimization by Flash Attention, the increase in sequence length does not make the inference time unacceptable.

## B  LOSS

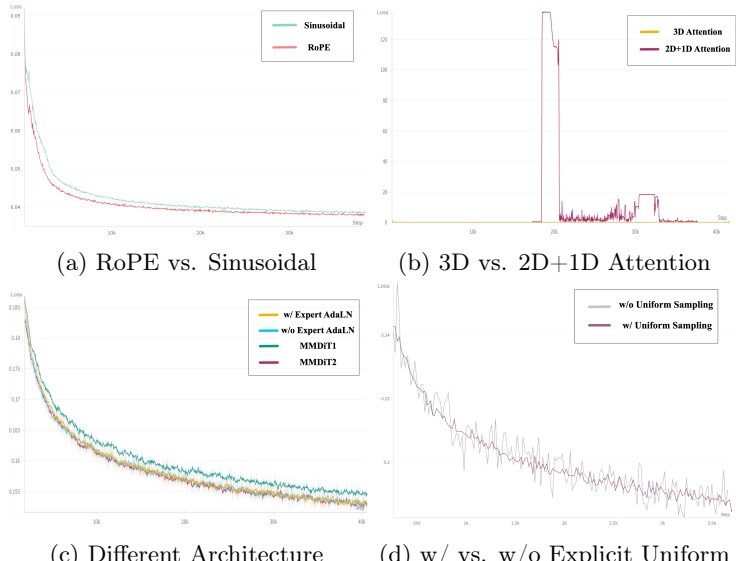

(a) RoPE vs. Sinusoidal   (b) 3D vs. 2D+1D Attention

(c) Different Architecture   (d) w/ vs. w/o Explicit Uniform

Figure 10: Training loss curve of different ablations.

Table 9: Validation loss at different diffusion timesteps when the training steps is 40k.

| Timestep | 100 | 300 | 500 | 700 | 900 |
|---|---|---|---|---|---|
| w/o explicit uniform sampling | 0.222 | 0.130 | 0.119 | 0.133 | 0.161 |
| w/ explicit uniform sampling | **0.216** | **0.126** | **0.116** | **0.129** | **0.157** |

## C  MORE EXAMPLES

More text-to-video examples are shown in Figure 11 and Figure 12.

## D  IMAGE TO VIDEO MODEL

We finetune an image-to-video model from the text-to-video model. Drawing from the (Blattmann et al., 2023), we add an image as an additional condition alongside the text. The image is passed through 3D VAE and concatenated with the noised input in the channel dimension. Similar to super-resolution tasks, there is a significant distribution gap between training and inference (the first frame of videos vs. real-world images). To enhance the model's robustness, we add large noise to the image condition during training. Some examples are shown in Figure 13, Figure 14. CogVideoX can handle different styles of image input.

## E  RELATED WORKS

**Video diffusion models**  Generating videos has been explored through various types of generative models, such as Generative Adversarial Networks (GANs) (Yu et al., 2022; Tulyakov et al., 2018), autoregressive methods (Hong et al., 2022; Yan et al., 2021), and non-autoregressive methods (Villegas et al., 2022; Yu et al., 2023a). Diffusion models have recently gained significant attention, achieving remarkable results in both image generation(Rombach et al., 2022; Esser et al., 2024) and video generation(Singer et al., 2022; Blattmann et al., 2023; Guo et al., 2023). However, the limited compression ratio and simple training strategy often restrict the generation to low-resolution short-duration videos (2-3 seconds), requiring multiple super-resolution and frame interpolation models to be cascaded(Singer et al., 2022; Ho et al., 2022) for a generation. This leads to generated videos with limited semantic information and minimal motion.

**Video VAEs**  To increase the compression ratio of videos and reduce computation costs, a common approach is to encode the video into a latent space using a Variational Autoencoder(VAE), which is also widely used in image generation. Early video models usually directly use image VAE for generation. However, modeling only the space dimension can result in jittery videos. SVD(Blattmann et al., 2023) tries to finetune the image VAE decoder to solve the jittering issue. However, this approach cannot take advantage of the temporal redundancy in videos and still cannot achieve an optimal compression rate. Recently, some video models(Zheng et al., 2024b; Lab & etc., 2024) try to use 3D VAE for temporal compression, but small latent channels still result in blurry and jittery videos.

**Text Prompt:** A few golden retrievers playing in the snow

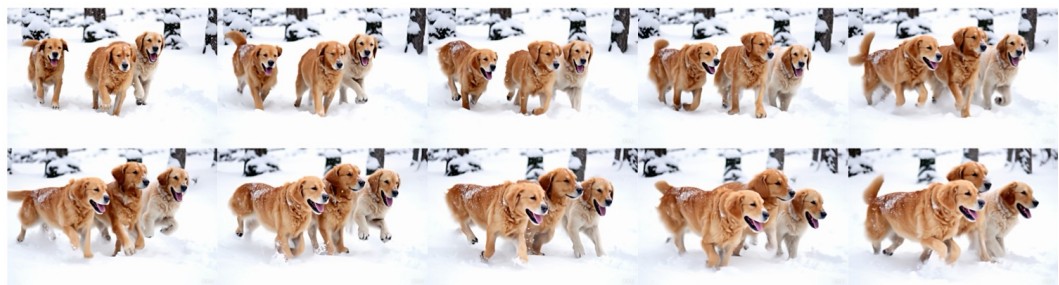

**Text Prompt:** A golden retriever with black sunglasses and long hair, with a rainy rooftop in the background, runs towards the camera from far to near

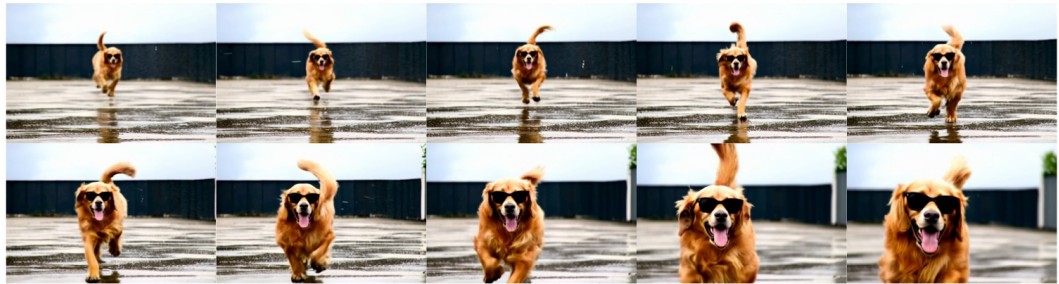

**Text Prompt:** Three dolphins leap out of the ocean at sunset, then splash into the water

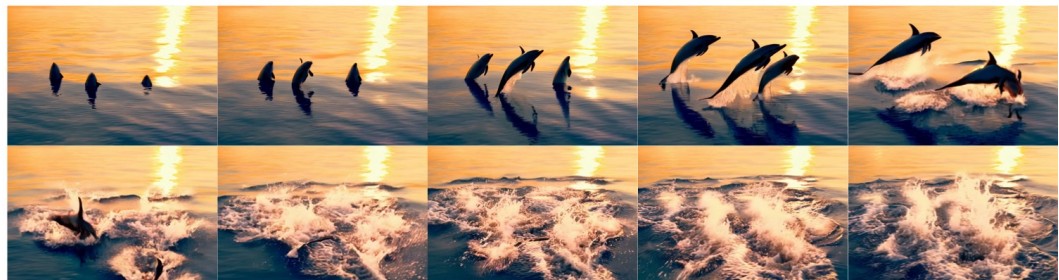

**Text Prompt:** The camera rotates around a stack of vintage televisions that show a variety of programs - 1950s sci-fi movies, horror movies, news, stills, 1970s sitcoms, etc. - set in a large gallery at the New York Museum.

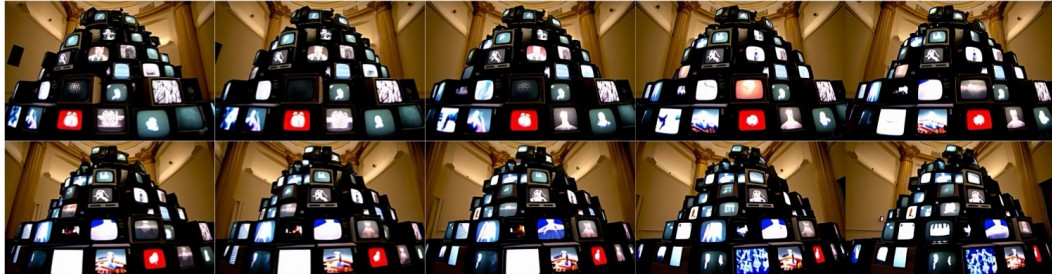

**Text Prompt:** Mushroom turns into a bear

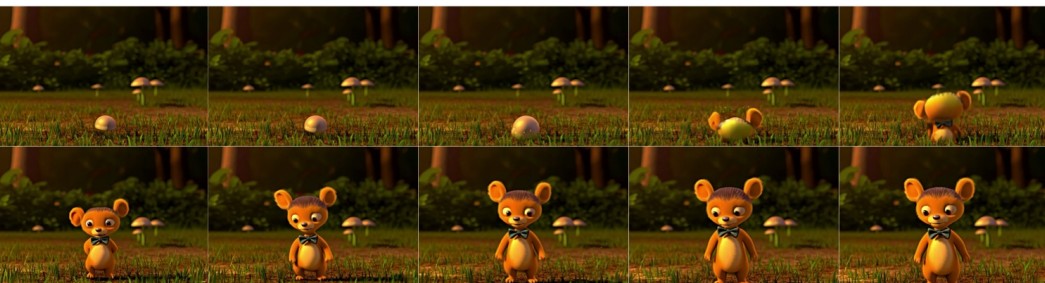

Figure 11: Text to video showcases. The displayed prompt will be upsampled before being fed into the model. The generated videos contain large motion and various styles.

**Text Prompt:** Push upward at a low angle, slowly look up, an evil dragon suddenly appears on the iceberg, and then the dragon spots you and rushes towards you. Hollywood movie style

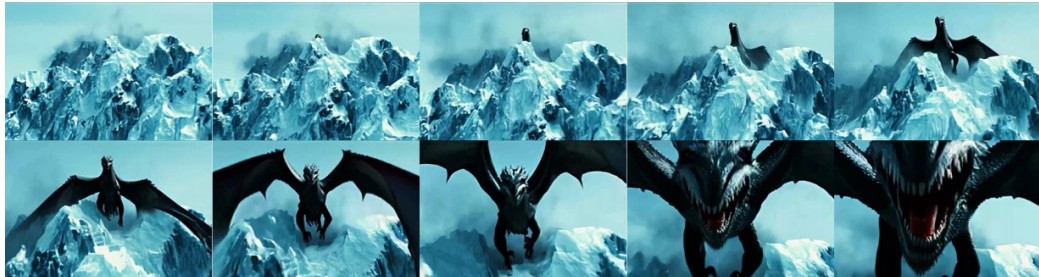

**Text Prompt:** An old-fashioned automobile drives through the streets of the Republic. While driving right in the middle of it, bombs suddenly fall from the sky, the car is blown up, the people in the car are blown up, the screen shakes, the movie winds up

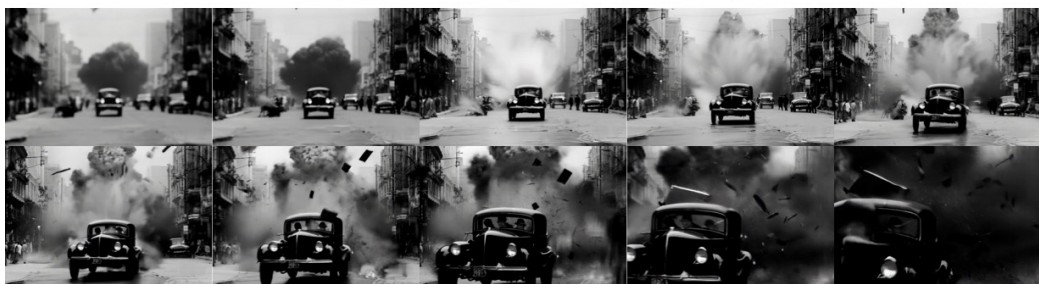

**Text Prompt:** A man running in the snow

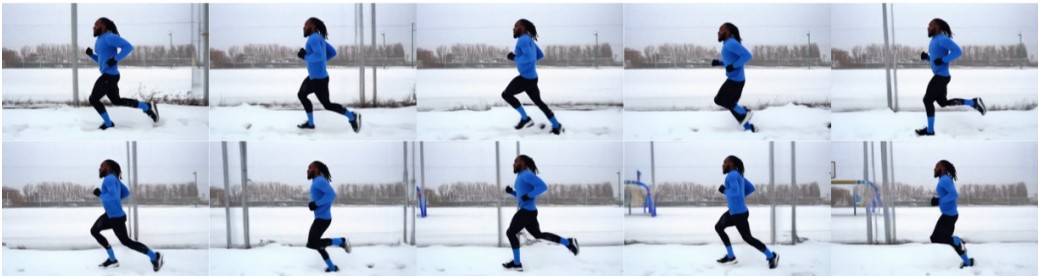

**Text Prompt:** The camera follows behind a white vintage SUV with a black roof rack as it speeds up a steep dirt road surrounded by pine trees on a steep mountain slope, dust kicks up from it's tires, the sunlight shines on the SUV as it speeds along the dirt road, casting a warm glow over the scene. The dirt road curves gently into the distance, with no other cars or vehicles in sight. The trees on either side of the road are redwoods, with patches of greenery scattered throughout. The car is seen from the rear following the curve with ease, making it seem as if it is on a rugged drive through the rugged terrain. The dirt road itself is surrounded by steep hills and mountains, with a clear blue sky above with wispy clouds.

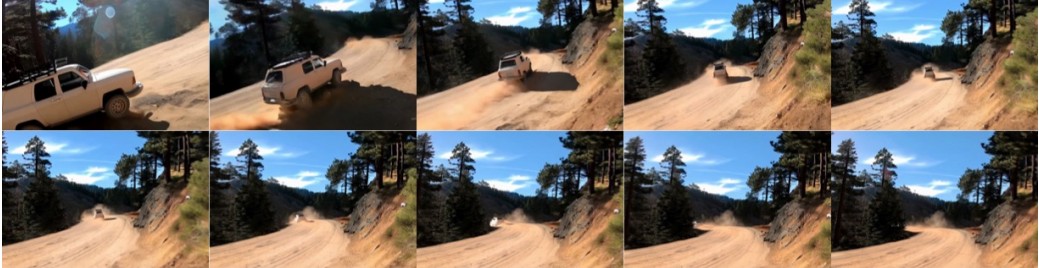

**Text Prompt:** In the cafe by the window, a man in a suit sits at the table and slowly raises his coffee to sip it, his eyes looking out the window, the street is full of traffic, the man is in deep thought.

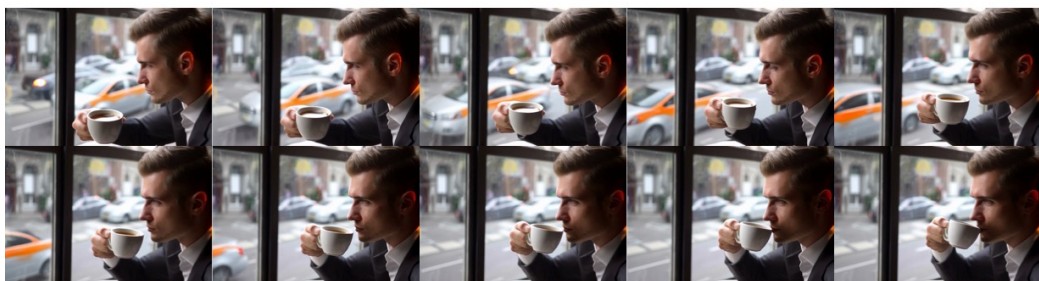

Figure 12: Text to video showcases.

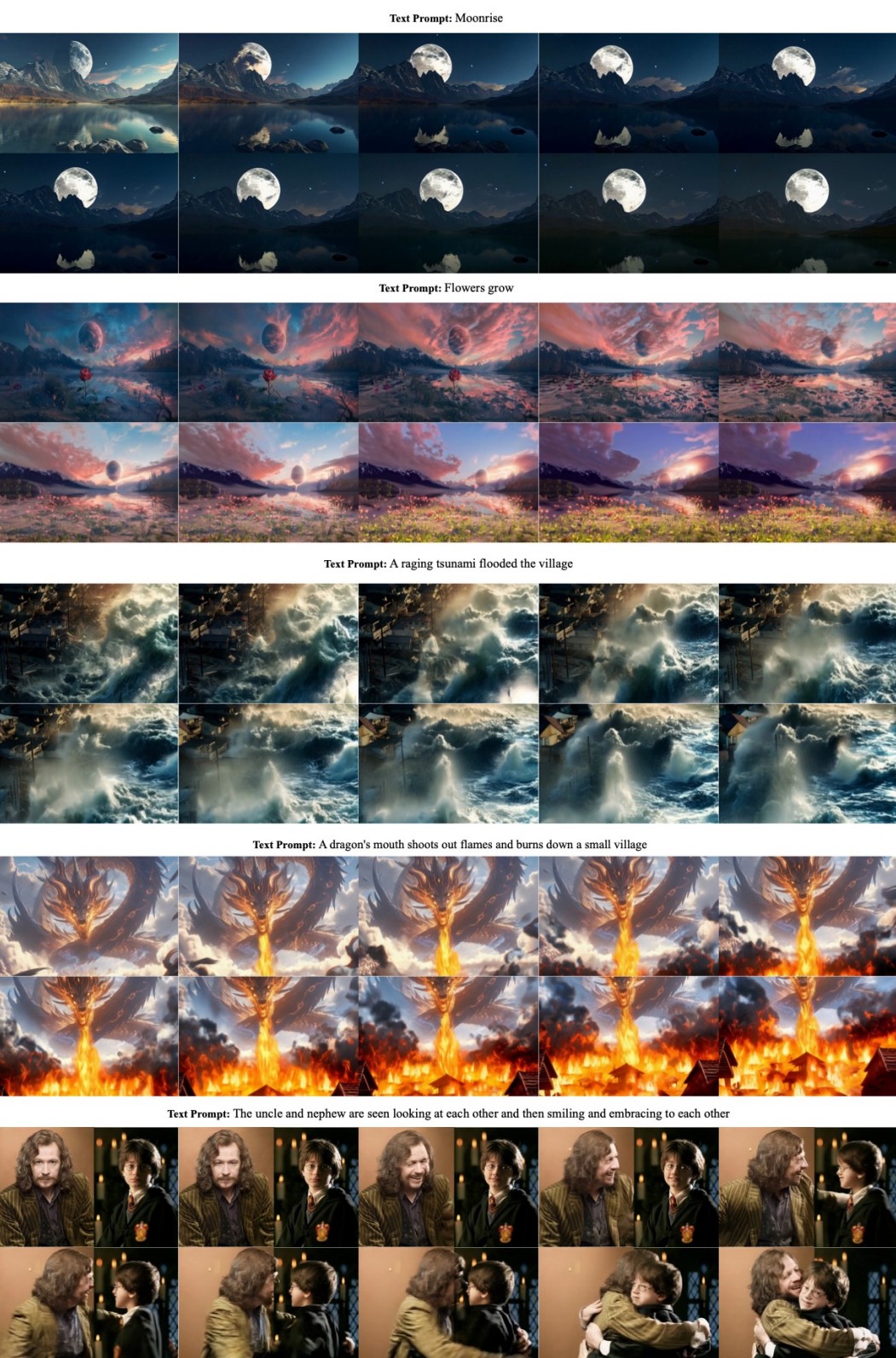

Figure 13: Image to video showcases. The displayed prompt will be upsampled before being fed into the model.

**Text Prompt:** An elephant slowly walks out of a cloud of fog, the fog shatters and flows

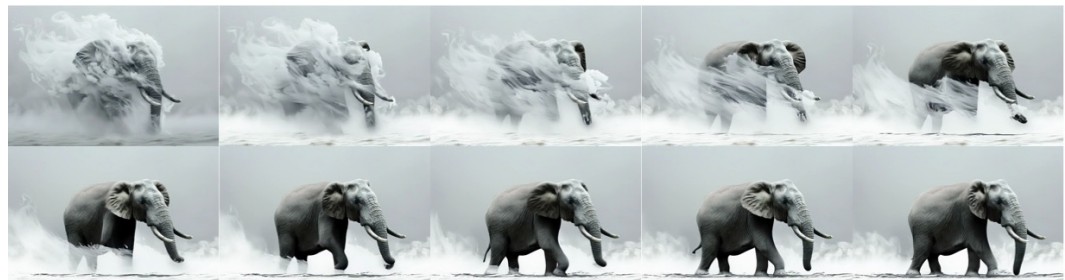

**Text Prompt:** A cat walks through water, breaking the surface, splashing water, light particles randomly fly, flowers and leaves sway

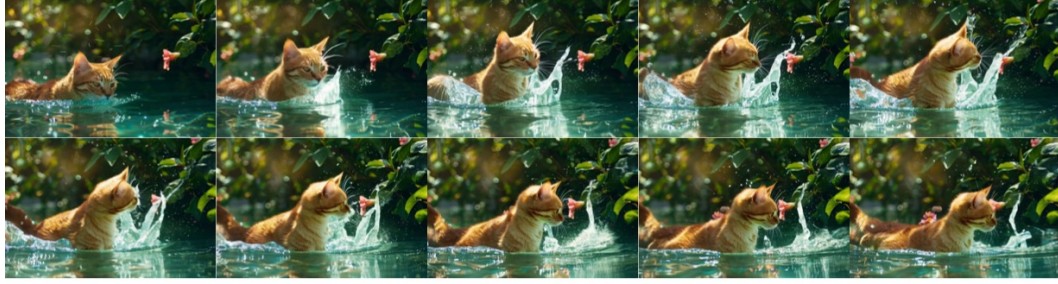

**Text Prompt:** A girl lowers her head and rubs her face against a puppy, the puppy looks up at the girl

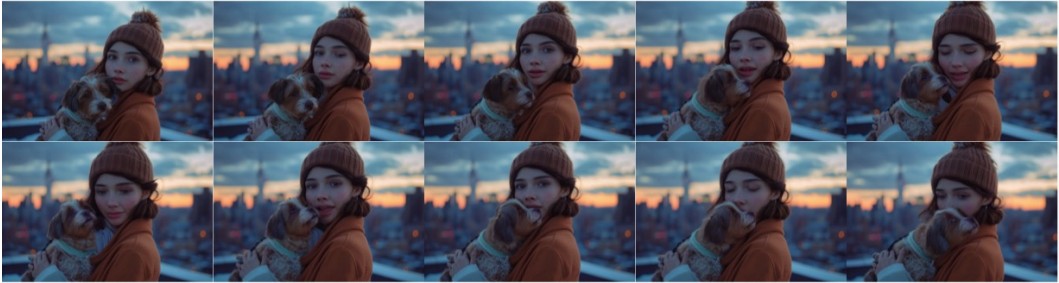

**Text Prompt:** A woman presses a camera shutter, her hair flying

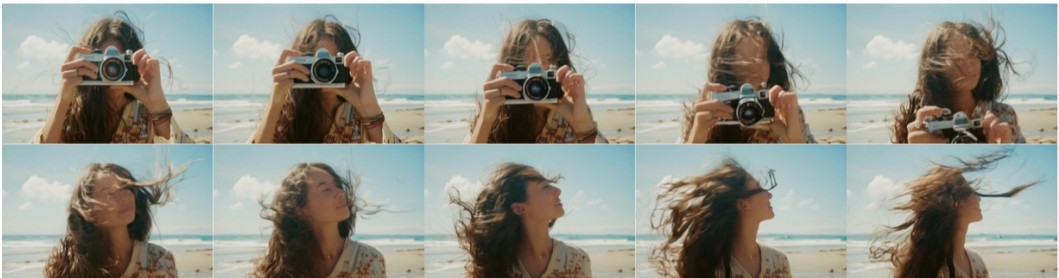

**Text Prompt:** A puppy closes its eyes, opens its mouth, and turns its head to bark

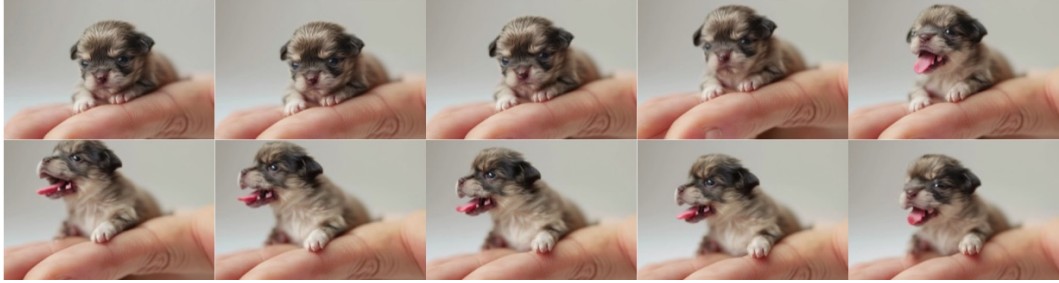

Figure 14: Image to video showcases.

## F    CAPTION UPSAMPLER

To ensure that text input distribution during inference is as close as possible to the distribution during training, similar to (Betker et al., 2023), we use a large language model to upsample the user's input during inference, making it more detailed and precise. Finetuned LLM can generate better prompts than zero/few-shot.

For image-to-video, we use the vision language model to upsample the prompt, such as GPT4V, CogVLM(Wang et al., 2023a).

---

**Zero-shot prompt for Text Upsampler**

```
You are part of a team of bots that create videos. You work
with an assistant bot that will draw anything you say in
square brackets. For example, outputting \" a beautiful
morning in the woods with the sun peaking through the
trees \" will trigger your partner bot to output a video
of a forest morning, as described. You will be prompted
by people looking to create detailed, amazing videos.
The way to accomplish this is to take their short prompts
and make them extremely detailed and descriptive.
There are a few rules to follow :
You will only ever output a single video description
per user request.
When modifications are requested, you should not simply
make the description longer. You should refactor the
entire description to integrate the suggestions.
```

---

## G    DENSE VIDEO CAPTION DATA GENERATION

In the pipeline for generating video captions, we extract one frame every two seconds for image captioning. Ultimately, we collected 50,000 data points to fine-tune the summary model. Below is the prompt we used for summarization with GPT-4:

---

**Prompt for GPT-4 Summary**

```
We extracted several frames from this video and described
each frame using an image understanding model,  stored
in the dictionary variable 'image_captions: Dict[str: str]'.
In 'image_captions',  the key is the second at which the image
appears in the  video,  and the value is a detailed description
of the image at that moment. Please describe the content of
this video  in as much detail as possible,  based on the
information  provided by 'image_captions',  including
the objects, scenery, animals, characters, and camera
movements within the video. \n image_captions={new_captions}\n
You should output your summary directly,  and not mention
variables like 'image_captions' in your response.
Do not include '\\n' and the word 'video' in your response.
Do not use introductory phrases such as: \"The video
presents\", \"The video depicts\", \"This video showcases\",
\"The video captures\" and so on.\n Please start the
description with the video content directly, such as \"A man
first sits in a chair, then stands up and walks to the
kitchen....\"\n Do not use phrases like: \"as the video
progressed\" and \"Throughout the video\".\n Please describe
```

---

```
the content of the video and the changes that occur, in
chronological order.\n Please keep the description of this
video within 100 English words.
```

## H    VIDEO CAPTION EXAMPLE

Below we present more examples to compare the performance of the Panda-70M video
captioning model and our CogVLM2-Caption model:

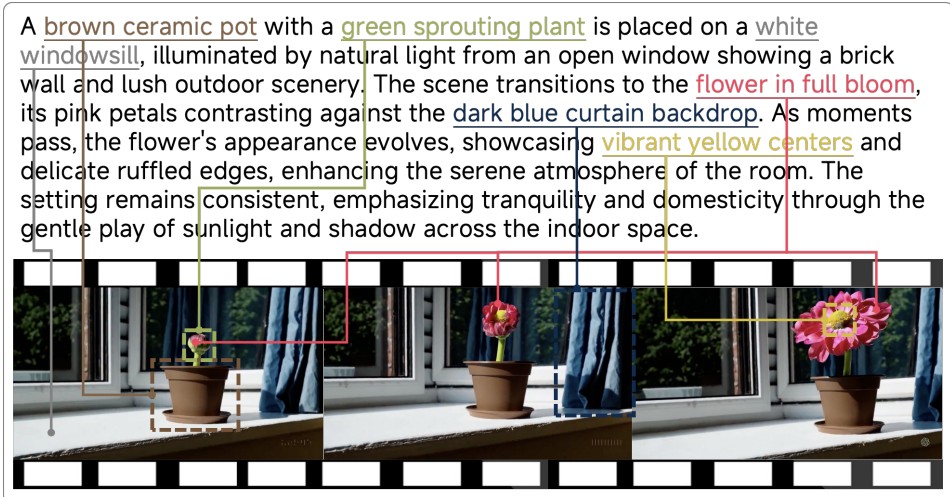

A brown ceramic pot with a green sprouting plant is placed on a white windowsill, illuminated by natural light from an open window showing a brick wall and lush outdoor scenery. The scene transitions to the flower in full bloom, its pink petals contrasting against the dark blue curtain backdrop. As moments pass, the flower's appearance evolves, showcasing vibrant yellow centers and delicate ruffled edges, enhancing the serene atmosphere of the room. The setting remains consistent, emphasizing tranquility and domesticity through the gentle play of sunlight and shadow across the indoor space.

Figure 15: An example from CogVLM2-Caption provides a detailed description of all specific objects and movements.

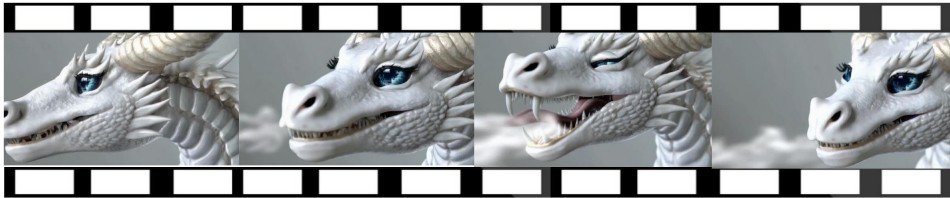

| **Caption Generated by Panda-70M** |
|---|
| `A close up of a white dragon with blue eyes.` |

| **Caption Generated by CogVLM2-Caption** |
|---|
| `A meticulously crafted white dragon with a serene expression`
`and piercing blue eyes is showcased in various close-ups, its`
`scales detailed with golden highlights. Initially, the dragon's`
`horns are curved back, but as time passes, they change to`
`spiraling gold, adding elegance. The creature's demeanor shifts`
`from tranquil majesty to a more formidable presence, with its`
`mouth agape revealing sharp teeth at times. Its expressions`
`alternate between calmness and fierce determination,`
`maintaining an air of mythical serenity throughout. The`
`consistent neutral background focuses attention on the dragon's` |

textured features and expressive eyes.

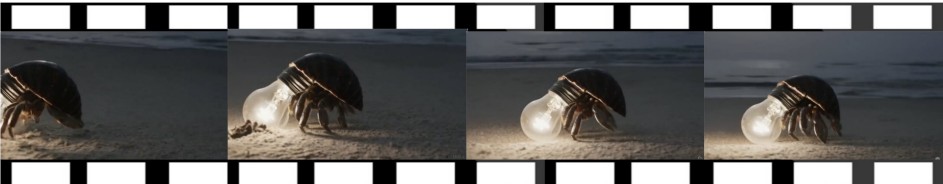

**Caption Generated by Panda-70M**

A crab is walking on the beach with a light bulb on its back.

**Caption Generated by CogVLM2-Caption**

A hermit crab with a dark, glossy shell and reddish-brown legs is seen carrying an illuminated light bulb on its back across the sandy terrain of a beach at night. The scene transitions from a soft glow to a more pronounced illumination as the crab moves, highlighting its delicate limbs against the smooth sand and tranquil sea backdrop. This surreal tableau blends natural beauty with human ingenuity, creating a serene yet whimsical atmosphere that emphasizes the crab's unique adaptation and the contrast between nature and technology in this quiet nocturnal setting.

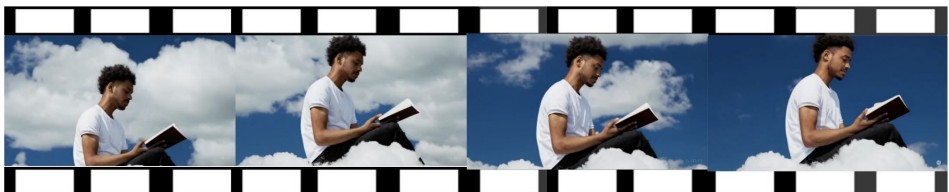

**Caption Generated by Panda-70M**

A young black man is sitting on a cloud and reading a book with a blue sky in the background.

**Caption Generated by CogVLM2-Caption**

A young Black man with an afro hairstyle and a neatly trimmed beard is seen sitting cross-legged on fluffy white clouds, deeply engrossed in reading a book with a red cover. He wears a plain white T-shirt and dark pants against a vivid blue sky dotted with cumulus clouds. Throughout the scenes, his expression remains one of deep concentration and peaceful contemplation, highlighting a moment of intellectual pursuit amidst nature's grandeur. The imagery suggests a serene atmosphere that emphasizes solitude and introspection, with no other people or objects around him.

# I  VIDEO TO VIDEO VIA COGVIDEOX AND COGVLM2-CAPTION

In this section, we present several examples of video-to-video generation using CogVideoX and CogVLM2-Caption. Specifically, we first input the original video into CogVLM2-Caption to obtain the video's caption, and then feed this caption into the CogVideoX model to generate a new video. From the examples below, it can be seen that our pipeline achieves a high degree of fidelity to the original video, showing that CogVLM2-Caption can capture almost all the details in the video.

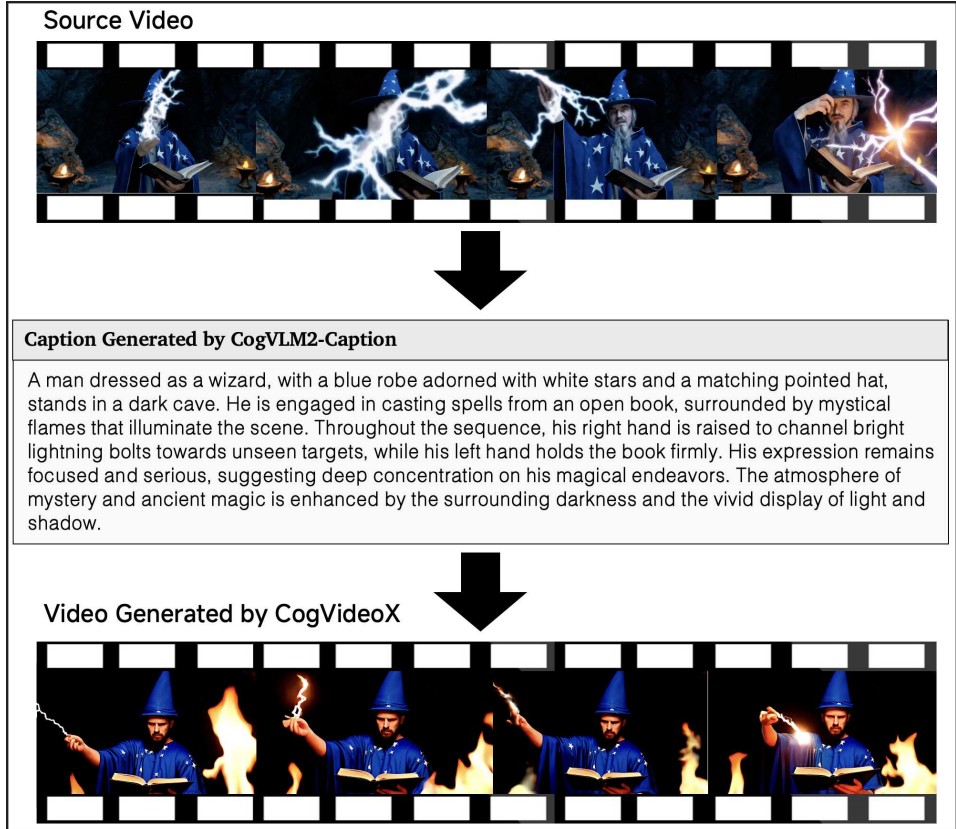

**Source Video**

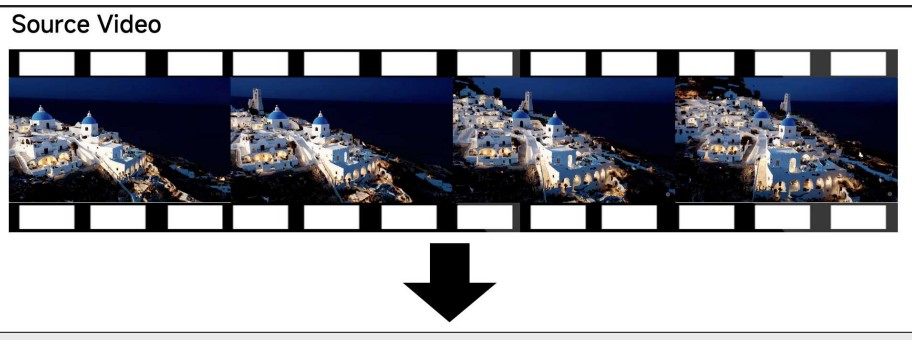

**Caption Generated by CogVLM2-Caption**

A picturesque evening descends on a cliffside village, showcasing whitewashed buildings with blue domes that glow against the darkening sky. The Aegean Sea mirrors this celestial hue, creating a serene tableau devoid of people and vehicles. As time passes, the scene remains tranquil, illuminated by golden lights from within homes and lit pathways weaving between structures. A solitary windmill stands out, symbolizing local culture amidst the peaceful setting. The absence of visible human activity emphasizes the stillness and beauty of the coastal hamlet, inviting contemplation in its embrace.

**Video Generated by CogVideoX**

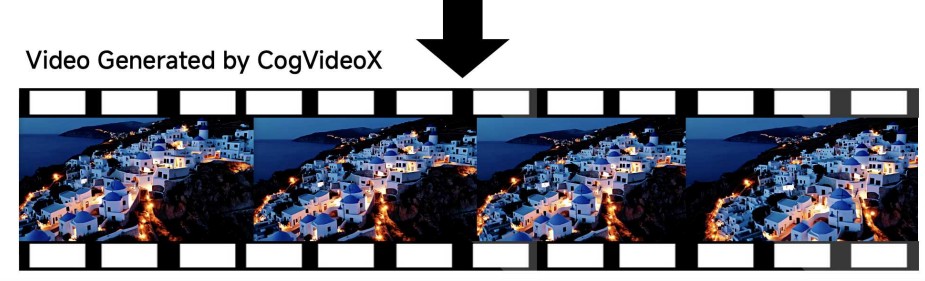

**Source Video**

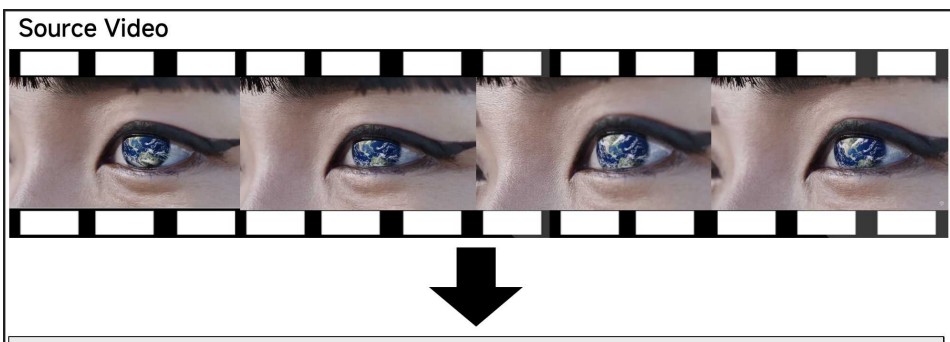

**Caption Generated by CogVLM2-Caption**

A woman's eye, in sharp focus and detailed with a bold black eyeliner, reflects the Earth. The vivid colors of blue oceans and green continents stand out against her clear iris, symbolizing a deep connection between humanity and our planet. Her expression remains neutral throughout, emphasizing introspection or awareness. As time passes, the reflection subtly shifts to include parts of Africa and Europe, suggesting a global perspective. The contrast between her dark eyelashes and light skin accentuates the visual metaphor for unity and interconnectedness, while her gaze suggests contemplation on environmental issues or a profound sense of responsibility towards the world.

**Video Generated by CogVideoX**

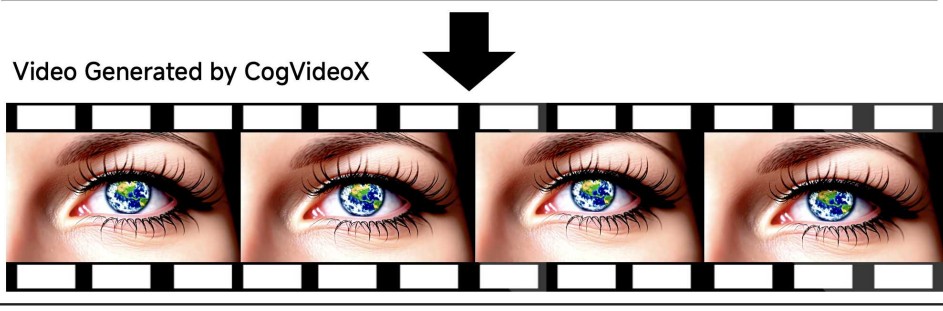

## J  HUMAN EVALUATION DETAILS

One hundred meticulously crafted prompts are used for human evaluators, characterized by their broad distribution, clear articulation, and well-defined conceptual scope.

A panel of evaluators is instructed to assign scores for each detail on a scale from zero to one, with the overall total score rated on a scale from 0 to 5, where higher scores reflect better video quality.

To better complement automated evaluation, human evaluation emphasizes the instruction-following capability: the total score cannot exceed 2 if the generated video fails to follow the instructions.

**Sensory Quality**: This part focuses mainly on the perceptual quality of videos, including subject consistency, frame continuity, and stability.

Table 10: Sensory Quality Evaluation Criteria.

| Score | Evaluation Criteria |
|---|---|
| 1 | High sensory quality: 1. The appearance and morphological features of objects in the video are completely consistent 2. High picture stability, maintaining high resolution consistently 3. Overall composition/color/boundaries match reality 4. The picture is visually appealing |
| 0.5 | Average sensory quality: 1. The appearance and morphological features of objects in the video are at least 80% consistent 2. Moderate picture stability, with only 50% of the frames maintaining high resolution 3. Overall composition/color/boundaries match reality by at least 70% 4. The picture has some visual appeal |
| 0 | Poor sensory quality: large inconsistencies in appearance and morphology, low video resolution, and composition/layout not matching reality |

**Instruction Following**: This part focuses on whether the generated video aligns with the prompt, including the accuracy of the subject, quantity, elements, and details.

Table 11: Instruction Following Evaluation Criteria.

| Score | Evaluation Criteria |
|---|---|
| 1 | 100% follow the text instruction requirements, including but not limited to: elements completely correct, quantity requirements consistent, elements complete, features accurate, etc. |
| 0.5 | 100% follow the text instruction requirements, but the implementation has minor flaws such as distorted main subjects or inaccurate features. |
| 0 | Does not 100% follow the text instruction requirements, with any of the following issues: 1. Generated elements are inaccurate 2. Quantity is incorrect 3. Elements are incomplete 4. Features are inaccurate |

**Physics Simulation**: This part focuses on whether the model can adhere to the objective law of the physical world, such as the lighting effect, interactions between different objects, and the realism of fluid dynamics.

Table 12: Physics Simulation Evaluation Criteria.

| Score | Evaluation Criteria |
|---|---|
| 1 | Good physical realism simulation capability, can achieve: 1. Real-time tracking 2. Good action understanding, ensuring dynamic realism of entities 3. Realistic lighting and shadow effects, high interaction fidelity 4. Accurate simulation of fluid motion |
| 0.5 | Average physical realism simulation capability, with some degradation in real-time tracking, dynamic realism, lighting and shadow effects, and fluid motion simulation. Issues include: 1. Slightly unnatural transitions in dynamic effects, with some discontinuities 2. Lighting and shadow effects not matching reality 3. Distorted interactions between objects 4. Floating fluid motion, not matching reality |
| 0 | Poor physical realism simulation capability, results do not match reality, obviously fake |

**Cover Quality**: This part mainly focuses on metrics that can be assessed from single-frame images, including aesthetic quality, clarity, and fidelity.

Table 13: Cover Quality Evaluation Criteria.

| Score | Evaluation Criteria |
|---|---|
| 1 | Image is clear, subject is obvious, display is complete, color tone is normal. |
| 0.5 | Image quality is average. The subject is relatively complete, color tone is normal. |
| 0 | Cover image resolution is low, image is blurry. |

## K  DATA FILTERING DETAILS

In order to obtain high-quality training data, we designed a set of negative labels to filter out low-quality data. Figure 16 presents our negative labels along with sample videos for each label.In table 14, we present the accuracy and recall of our classifier, trained based on video-llama, on the test set (10% randomly labeled data).

Table 14: Summary of Classifiers Performance on the Test Set. TP: True Positive, FP: False Positive, TN: True Negative, FN: False Negative.

| Classifier | TP | FP | TN | FN | Test Acc |
|---|---|---|---|---|---|
| Classifier - Editing | 0.81 | 0.02 | 0.09 | 0.08 | 0.91 |
| Classifier - Static | 0.48 | 0.04 | 0.44 | 0.04 | 0.92 |
| Classifier - Lecture | 0.52 | 0.00 | 0.47 | 0.01 | 0.99 |
| Classifier - Text | 0.60 | 0.03 | 0.36 | 0.02 | 0.96 |
| Classifier - Screenshot | 0.61 | 0.01 | 0.37 | 0.01 | 0.98 |
| Classifier - Low Quality | 0.80 | 0.02 | 0.09 | 0.09 | 0.89 |

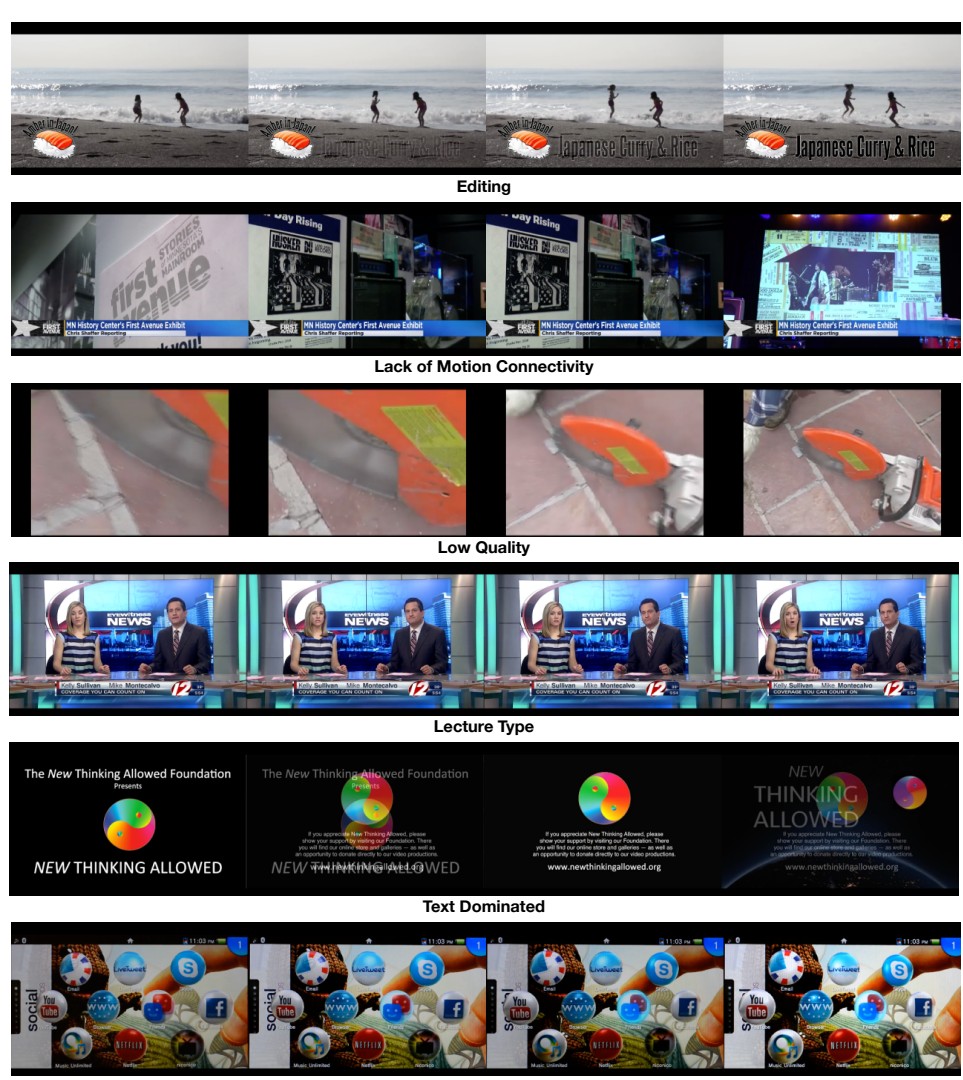

Figure 16: Examples of negative labels for video filtering.

