# OpenReview forum: "CogVideoX: Text-to-Video Diffusion Models with An Expert Transformer"
_ICLR.cc/2025/Conference — ICLR 2025 Poster_

### Official Review · Reviewer_YY2H · 2024-10-29

**Soundness:** 4
**Presentation:** 3
**Contribution:** 4
**Rating:** 6
**Confidence:** 4

**Summary:**

CogVideoX is a large-scale text-to-video generation model using a diffusion transformer, capable of generating 10-second videos at 16 fps with a resolution of 768×1360 pixels. It addresses previous models' limitations in movement and duration by introducing three components: a 3D Variational Autoencoder for improved video compression and fidelity, an expert transformer with adaptive LayerNorm for enhanced text-video alignment, and progressive training with multi-resolution frame techniques for coherent, long-duration videos with significant motions. The paper introduces a text-video data processing pipeline to enhance generation quality and semantic alignment. CogVideoX achieves state-of-the-art performance in text-to-video generation, and its components. The authors also make the model weights of the 3D Causal VAE, the video caption model, and CogVideoX publicly available.

**Strengths:**

1. The open-sourcing of the 3D Causal VAE, the video caption model, and the CogVideoX model significantly promotes future developments in video generation research.
2. The method can generate videos with larger frames, better temporal consistency, and higher resolution.
3. The proposed Multi-Resolution Frame Pack and Progressive Training techniques are interesting and meaningful.
4. This paper introduces sufficient technical improvements to enhance the model's performance.
5. Experimental results demonstrate that CogVideoX achieves superior performance compared to current text-to-video generation models.

**Weaknesses:**

1. Providing additional details in the methods section would enhance the paper’s completeness. For example, further explanation on implementing videos of different durations (and resolutions) in the same batch would be helpful. Although it is mentioned that the method is inspired by Patch n’Pack, a brief clarification would be beneficial.
2. It would be better to discuss the additional computational costs associated with using 3D attention compared to the commonly used 2D+1D attention.
3. The axis labels in Figure 8 appear to be mislabeled.
4. It would be better to add a related work section to discuss video generation works and the differences from previous methods and architectures. Additionally, references for some methods are missing in Table 3.

**Questions:**

1. There is a typo in line 310: "Patch’n Pack."
2. Line 459 seems to be missing a citation.

---

> ### Author Response · Authors · 2024-11-21
>
> Thanks for your valuable suggestions! We have provided explanations and additional experiments to address the weaknesses.
>
> **Weakness 1.Providing additional details in the methods section would enhance the paper’s completeness**
>
>  We explained our motivation for choosing this method in lines 296–392, as follows:
>   1. For resolutions, direct cropping can result in incomplete subjects in the generated videos, as demonstrated in Sora. And cannot efficiently utilize data from all resolutions for training. The same principle applies to video duration. As illustrated in Figure 3 of the paper, directly cropping video duration results in incomplete narratives and inefficient data utilization. This issue has already been explained in SDXL[1] and Sora[2], so we did not include it in our paper. However, we added citations in the corresponding part of the paper.
>   2. Another issue is the significant distribution gap between images and videos during training. This causes the model to treat image generation and fixed-duration video generation as two separate tasks, leading to poor generalization. Mixed-duration training can help mitigate this problem.
>   4. Another advantage is that this packing strategy supports generating videos with various aspect ratios and duration without the need for complex data bucketing operations like SDXL.
>   We have added those explanations to the newly uploaded pdf.
> [1] Podell, Dustin, et al. "Sdxl: Improving latent diffusion models for high-resolution image synthesis." arXiv preprint arXiv:2307.01952 (2023).
> [2]https://openai.com/index/sora/
>
> **Weakness 2. It would be better to discuss the additional computational costs associated with using 3D attention compared to the commonly used 2D+1D attention.**
>
> | Method   | 256×384×6s | 480×720×6s | 768×1360×5s |
> |----------|-------------|------------|-------------|
> | 2D+1D    | 0.38s       | 1.26s      | 4.17s       |
> | 3D       | 0.41s       | 2.11s      | 9.60s       |
>
> We added a time comparison table and included it in Appendix A. As shown in the table above, the computational cost of 3D attention is indeed larger than 2D+1D attention when the sequence gets longer. However, considering the shortcomings of 2D+1D attention in the experiment, we still choose 3D attention.
>
> **Weakness 3. The axis labels in Figure 8 appear to be mislabeled.**
>
> Thanks for pointing that out. We have fixed it in the newly uploaded PDF.
>
> **Weakness 4. It would be better to add a related work section to discuss video generation works and the differences from previous methods and architectures. Additionally, references for some methods are missing in Table 3**
>
> Considering the page limitation, we have added a "Related Works" section in the appendix and added references in Table 3.

---

> > ### Comment · Reviewer_YY2H · 2024-11-24
> >
> > Thanks to the authors for addressing my concerns in the rebuttal, I will keep my original score.

---

> > > ### Author Response · Authors · 2024-11-29
> > >
> > > Thank you for your recognition, we will continue to work hard to promote the development of this field

---

### Official Review · Reviewer_xLCm · 2024-11-01

**Soundness:** 3
**Presentation:** 3
**Contribution:** 3
**Rating:** 6
**Confidence:** 4

**Summary:**

The authors introduced CogVideoX, a large-scale text-to-video generation model based on diffusion transformers, addressing common issues in prior video generation models, such as limited motion and short duration. It introduces several designs like 3D VAE, which I think is novel.

Experimental results demonstrate that CogVideoX achieves state-of-the-art performance across multiple machine metrics and human evaluations.

**Strengths:**

1.There are few diffusion transformer-based models. This paper provides a comprehensive approach, from training a 3D VAE to constructing a text-video data processing pipeline, along with a robust model architecture and training design. The model and code are well-implemented.

2.The paper is well-structured, and very easy to follow.

3.The ablation study is thorough, verifying the effects of 2D+1D attention and 3D attention design in video generation, as well as different positional encodings. The 2B and 5B model training partially validates the scaling law.

**Weaknesses:**

1. The novelty at the model level is relatively weak, with two expert adaptive layer norms being a fairly simple design.
2. The authors use T5 as the encoder, but comparisons with mixed text encoders are missing.
3. The 3D VAE design is primarily compared with its own variants, without comparison to other VAE designs, such as the spatial compression capability of SD’s 2D VAE.
4.Although the data processing pipeline is clearly outlined, the dataset is not publicly available.  It would be nice to open-source the data.

**Questions:**

None

---

> ### Author Response · Authors · 2024-11-21
>
> Thanks for your valuable suggestions! We have provided explanations and additional experiments to address the weaknesses.
>
> **Weakness 1. The novelty at the model level is relatively weak, with two expert adaptive layer norms being a fairly simple design.**
>
> Besides expert adaptive layernorm, we propose 3D rope, which decompose the original 1D rope into three dimensions: time, height, and width, helpful to model the video information. And we reveal the inherent shortcomings of 2D+1D attention in modeling video information and confirm the importance of 3D attention.
>
> **Weakness 2. The authors use T5 as the encoder, but comparisons with mixed text encoders are missing.**
>
> Mixed text encoder is a known trick for imporoving model performance[1][2][3], and we believe it will work. However, text encoder selecting is not the main focus of this work, and it would increase the complexity of the model architecture. Some works have already shown that a standalone T5 encoder is sufficient to achieve strong semantic alignment[3][4], which is why we chose to use T5 as the sole encoder.
>
> [1] Balaji, Yogesh, et al. "ediff-i: Text-to-image diffusion models with an ensemble of expert denoisers." arXiv preprint arXiv:2211.01324 (2022).
>
> [2] Podell, Dustin, et al. "Sdxl: Improving latent diffusion models for high-resolution image synthesis." arXiv preprint arXiv:2307.01952 (2023).
>
> [3] Esser, Patrick, et al. "Scaling rectified flow transformers for high-resolution image synthesis." Forty-first International Conference on Machine Learning. 2024.
>
> [4] Saharia, Chitwan, et al. "Photorealistic text-to-image diffusion models with deep language understanding." Advances in neural information processing systems 35 (2022): 36479-36494.
>
> **Weakness 3. The 3D VAE design is primarily compared with its own variants, without comparison to other VAE designs, such as the spatial compression capability of SD’s 2D VAE.**
>
>   1. We have compared with the SDXL VAE in Table 1, where PSNR is calculated for each frame and then averaged. Therefore, the PSNR comparison can demonstrate that our per-frame reconstruction capability on video is superior to that of the SDXL VAE. Additionally, in Table 2, we also compared with the 3DVAE applied in other video models (Open-Sora, Open-Sora-Plan).
>
>   2. For a more intuitive comparison, we add new visual comparisons for different VAE on the demo website (https://cogvideox4iclr.github.io/cogvideox-demo/).  We will also update these examples in the appendix later. It can be observed that other 3D VAEs often result in blurry outputs, while 2D VAEs like SDXL exhibit noticeable jittering. Our VAE effectively addresses both issues.
>
> **Weakness 4. Although the data processing pipeline is clearly outlined, the dataset is not publicly available. It would be nice to open-source the data.**
>
> The primary source of our data is YouTube Videos (with same URLs as pandas70M). We believe that training on this panda70M alone can achieve similar results. Additionally, to further enhance model performance, we utilized a samller in-house dataset, which is not feasible to open-source now due to company policies.

---

### Official Review · Reviewer_Y5bt · 2024-11-01

**Soundness:** 2
**Presentation:** 3
**Contribution:** 3
**Rating:** 8
**Confidence:** 5

**Summary:**

This paper proposes a text-to-video generation model focused on generating video with temporal consistency and rich motion in longer sequence. It proposes a 3D VAE for video compression, achieving high-quality reconstruction, and introduces a new Transformer architecture to enhance semantic alignment between text and video. Experimental results show that CogVideoX outperforms existing models, especially in complex dynamic scenes. With open-source release, CogVideoX has potential to advance research in video generation.

**Strengths:**

* Easy to follow
* Effective apporach to text-to-video generation based on diffusion transformer
* Demonstration of high-quality video

**Weaknesses:**

* Overall, the performance improvement does not appear to be significant. For example, the CogVideoX-2B model outperforms only in the Dynamic Degree (compared to CogVideoX-5B). Additionally, CogVideoX-5B does not achieve the best performance across all models and metrics.
* The computational cost (in both time and memory) and the complexity of data filtering and training seem high. Authors should specify these.

**Questions:**

* In the paper, ablation studies have been evaluated with only FVD scores. However, for Expert AdaLN, which focuses on alignment between text and video data, it would be reasonable to include other metrics, such as the CLIP score, to provide more robust validation of the ablation study results.

* What causes the 2B model to underperform compared to other baselines and the 5B model except dynamic degree?

* Can the model generate similar-quality videos without using the caption upsampler? The caption upsampler may hinder robust generalization performance.

---

> ### Author Response · Authors · 2024-11-21
>
> Thanks for your valuable suggestions! We have provided explanations and additional experiments to address the weaknesses.
>
> **Weakness 1 and Question 2: the performance improvement does not appear to be significant. For example, the CogVideoX-2B model outperforms only in the Dynamic Degree (compared to CogVideoX-5B).**
>
> We sincerely apologize for the previous mistake where we inadvertently used the sampling settings for the 5B model during the evaluation of the 2B model. Due to slight differences in the training processes of the two models, their respective optimal sampling settings should differ as well. We have re-evaluated the 2B model and updated the score in the pdf.
>
> Dynamic degree is used to measure the level of motion in generated videos, but it is unrelated to the visual quality of the generation. It can be observed that the dynamic degree of both CogVideoX-2B and 5B are higher than others, indicating that our models can generate videos with large motions compared to other models. However, **maintaining visual quality under high dynamic motion and long durations is extremely challenging**, which highlights how the performance of the CogVideoX models surpasses that of the baseline models.
>
> The dynamic degree of the 2B model is **slightly higher** than that of the 5B model, which we believe is due to the occasional generation of distorted videos by the 2B model.
>
> **Weakness2: The computational cost (in both time and memory) and the complexity of data filtering and training seem high. Authors should specify these.**
>
>
> We have provided additional details on the time and spatial consumption corresponding to different resolutions in Appendix A, and referenced them in the main text.
>
> Use H800, bf16, 50steps to generate a video:
>
> |        | 2b-480x720-6s | 5b-480x720-6s | 2b-768x1360-5s | 5b-768x1360-5s |
> | ------ | ------------- | ------------- | -------------- | -------------- |
> | Time   | 49s           | 113s          | 220s           | 500s           |
> | Memory | 18GB          | 26GB          | 53GB           | 76GB           |
>
> **Question 1: In the paper, ablation studies have been evaluated with only FVD scores. However, for Expert AdaLN, which focuses on alignment between text and video data, it would be reasonable to include other metrics, such as the CLIP score, to provide more robust validation of the ablation study results.**
>
> Generally, the FVD similarity between two videos with similar caption is higher than that between two random videos. In our ablation, the FVD comparison is conducted betweend the ground truth and the generated videos corresponding to the same prompts. Therefore, FVD can, to some extent, measure the semantic alignment of video generation.
>
> But, we agree that more semantic comparisons should be added. We add CLIP4CLIP[1] results in **figure8** to compare the semantic similarity between videos. CLIP4CLIP is a video embedding model aligned with text finetuned from CLIP.  The semantic alignment of the CogVideoX is better than the MMDiT with twice the parameters and the structure without expert AdaLN.
>
> [1] Luo, Huaishao, et al. "Clip4clip: An empirical study of clip for end to end video clip retrieval and captioning." Neurocomputing 508 (2022): 293-304.
>
> **Question 3: Can the model generate similar-quality videos without using the caption upsampler? The caption upsampler may hinder robust generalization performance.**
>
>  The model can generate corresponding videos with short captions, but the quality will be slightly worse compared to longer captions.
>
> 1. **Why use long caption during training and inference?**  And long caption training and inference with an upsampler is widely adopted in image generation[2]. Performance improvement can be attributed to longer captions providing more accurate visual information, thereby reducing the difficulty of generation for the model and shifting this part of the challenge to the LLM. Due to the strong generalization capabilities of current LLMs, they can fully perform reasonable upsampling on all input prompts, without hindering the model's robustness.
>
> 2. **Short caption performance.** Since most captions during training are long, using short captions during inference leads to a training-inference mismatch, resulting in poorer generation quality. Ensuring that the model can generate results for short captions that are as effective as those for long captions is an intriguing challenge. However, it is not the focus of this paper, and we look forward to exploring it in future work.
>
> [2] Betker, James, et al. "Improving image generation with better captions." Computer Science. https://cdn.openai.com/papers/dall-e-3. pdf 2.3 (2023): 8.

---

> > ### Comment · Reviewer_Y5bt · 2024-11-23
> >
> > Thank you for your detailed response. My concerns have been partially addressed.
> >
> > Does the distorted video of the 2B model imply that the actions are jittering? It would be helpful to provide qualitative samples and comparisons to illustrate where such issues occur. If this issue is resolved, I will happily raise the score.

---

> > > ### Author Response · Authors · 2024-11-24
> > >
> > > We added some comparisons between 2B and 5B models to a new demo website.
> > >
> > > https://cogvideox4iclr.github.io/cogvideox-demo2/
> > >
> > > The 2B model sometimes generates objects that deform or noise during motion. The 5B model also encounters similar issues, but the occurrence rate is significantly lower. We believe this is primarily due to insufficient training time and model parameters. Sora has shown in the blog that continuing to scale might solve the problem.
> > >
> > > This problem also occurs in image generation, when the training and parameters are not enough, the model sometimes generates distorted human limbs.

---

> > > > ### Comment · Reviewer_Y5bt · 2024-11-25
> > > >
> > > > I appreciate to this comment. My concern is all addressed. I think your work can contribute enough to the research field. I'll raise the score to 8 (good paper).

---

> > > > > ### Author Response · Authors · 2024-11-29
> > > > >
> > > > > Thank you for your recognition, we will continue to work hard to promote the development of this field

---

### Official Review · Reviewer_QWeS · 2024-11-02

**Soundness:** 3
**Presentation:** 4
**Contribution:** 3
**Rating:** 8
**Confidence:** 4

**Summary:**

This paper is a text-to-video generation model using a diffusion transformer that generates 10-second, high-resolution videos aligned with text prompts without super-resolution or frame-interpolation. It improves upon previous models through a 3D causal VAE for text-video alignment, coherent, long-duration, and motion-rich videos. It achieves state-of-the-art performance and is open-source

**Strengths:**

I believe this paper is good, an open-source and open-hyperparameters, will have a impact on community research.

1. The paper is well written.
2. This paper creates a well-defined dataset for text-to-video generation.
3. This paper publicly release 5B and 2B models, including text-to-video and image-to-video versions.
4. This paper achieves state-of-the-art performance compared with other text-to-video models.
5. The generated high-resolution videos have very good quality.

**Weaknesses:**

1. The training process and model structure are somewhat unintuitive and slightly complex, raising some concerns about performance improvement in the future.
2. There is a lack of detailed analysis on the ablation study. If there were a more detailed analysis on the ablation study, I would raise the score.
3. Why is 2D + 1D attention unstable? In Figure 8, is the X-axis FVD and the Y-axis Training Steps?
4. In 459 line, () is typo ?
5. In 475 line, 17 frame is right ?

**Questions:**

Please refer to the weaknesses.

---

> ### Author Response · Authors · 2024-11-21
>
> Thanks for your valuable suggestions! We have provided explanations and additional experiments to address the weaknesses.
>
> **Weakness1. The training process and model structure are somewhat unintuitive and slightly complex, raising some concerns about performance improvement in the future.**
>
> 1. For model structure, the 2B and 5B model training partially validates the scaling law. Considering that the current SOTA image/video generation models are already at the scale of 20B to 30B parameters([1],[2]), we believe that further increasing the model parameters can continue to improve performance.
>
> 2. For the training process, some other works([2],[3]) have shown that multi-stage training can enhance training efficiency and generation quality, and it has already been widely adopted. Furthermore, the success of LLMs ([4]) also confirms the scalability and effectiveness of multi-stage training.
>
> [1] Liu, Bingchen, et al. "Playground v3: Improving text-to-image alignment with deep-fusion large language models." arXiv preprint arXiv:2409.10695 (2024).
>
> [2] Polyak, Adam, et al. "Movie gen: A cast of media foundation models." arXiv preprint arXiv:2410.13720 (2024).
>
> [3] Podell, Dustin, et al. "Sdxl: Improving latent diffusion models for high-resolution image synthesis." arXiv preprint arXiv:2307.01952 (2023).
>
> [4] Touvron, Hugo, et al. "Llama 2: Open foundation and fine-tuned chat models." arXiv preprint arXiv:2307.09288 (2023).
>
> **Weakness 2: There is a lack of detailed analysis on the ablation study. If there were a more detailed analysis on the ablation study, I would raise the score.**
>
> In the newly uploaded PDF, we have added more analysis in the ablation study section and included a comparison of CLIP4Clip scores (semantic alignment) between different model variants.
>
> **Weakness3: Why is 2D + 1D attention unstable? In Figure 8, is the X-axis FVD and the Y-axis Training Steps?**
>
>   1. First, as shown in Figure 5, 2D+1D attention requires implicit transmission of visual information,  increasing the learning complexity and making it challenging to maintain the consistency of large-movement objects. And we suppose that as the model size increases, such as 5B, training becomes more prone to instability, placing higher demands on the structural design. The 2D+1D structure, as discussed in section 2.2, is not suitable for video generation tasks, which may increase instability during training.
>
>   2. We have updated figure8 and can be clearly see in the figure that even in the early steps before 2d+1d model collapsed, its FVD is much higher than the 3D structure.
>
>   3. Thanks for pointing out the drawing error in figure8. We have fixed it in the newly uploaded PDF.
>
> **Weakness 4: In 459 line, () is typo ?**
>
> We have fixed it in the newly uploaded PDF.
>
> **Weakness 5: In 475 line, 17 frame is right ?**
>
> 17 frame is right. In our experiment, VAE is already able to effectively learn temporal modeling with 17 frames. And we also train the VAE with 129 frames to generalize to longer videos in the second stage. The purpose of the staged training is to improve training efficiency.

---

> > ### Comment · Reviewer_QWeS · 2024-11-24
> >
> > I reviewed the rebuttal and all comments from other reviewers. Thank you for your detailed responses.
> > My concerns have been addressed, and I am raising my score to 8(good paper).
> >
> > I believe the open-sourcing of this paper, along with its high-quality performance, will have a positive impact on the generative model research field.

---

> > > ### Author Response · Authors · 2024-11-29
> > >
> > > Thank you for your recognition, we will continue to work hard to promote the development of this field

---

### Official Review · Reviewer_DUFK · 2024-11-07

**Soundness:** 3
**Presentation:** 3
**Contribution:** 3
**Rating:** 6
**Confidence:** 2

**Summary:**

The paper introduces CogVideoX, an advanced text-to-video generation model built on a diffusion transformer. It produces 10-second, high-resolution videos (768×1360) at 16 fps. To overcome challenges in video generation, the authors implement:
A 3D Variational Autoencoder (VAE) for better compression and video quality.
An Expert Transformer with adaptive LayerNorm to enhance text-video alignment.
Progressive Training and Multi-Resolution Frame Packing to create coherent, long-duration videos with significant motion.

**Strengths:**

1. 3D-RoPE for Video Data: The adaptation of Rotary Position Embedding (RoPE) to 3D (3D-RoPE) is novel, effectively capturing spatiotemporal relationships and adding originality to positional encoding.
2. The qualitative visualizations showcase various video domains, including scenes, single-object videos, and multi-object videos.

**Weaknesses:**

1. The paper highlights CogVideoX’s performance but lacks a detailed analysis of computational efficiency, including memory usage and training/inference time.
2. The paper does not discuss the scalability of CogVideoX to longer video durations beyond 10 seconds.

**Questions:**

Please refer to the weakness part.

---

> ### Author Response · Authors · 2024-11-21
>
> Thanks for your valueable suggestions! We have provided explanations and additional experiments to address the weaknesses.
>
> **Weakness 1. The paper highlights CogVideoX’s performance but lacks a detailed analysis of computational efficiency, including memory usage and training/inference time.**
>
> We tested the time and memory comsumption of 2b and 5b model, which are listed below and in Appendix A.  The time consumption for other durations can be extrapolated by sequence.
>
> Use H800, bf16, 50steps to generate a video:
>
> |        | 2b-480x720-6s | 5b-480x720-6s | 2b-768x1360-5s | 5b-768x1360-5s |
> | ------ | ------------- | ------------- | -------------- | -------------- |
> | Time   | 49s           | 113s          | 220s           | 500s           |
> | Memory | 18GB          | 26GB          | 53GB           | 76GB           |
>
> As for training time: This is highly correlated with the number of GPUS and the pretrain infra, which can result in significant differences. Overall, we used more than 100,000 H100 hours during pretraining.
>
> **Weakness 2. The paper does not discuss the scalability of CogVideoX to longer video durations beyond 10 seconds.**
>
> Currently, almost all models capable of generating videos longer than 10 seconds rely on non-end-to-end methods[1][2], which can be directly applied to our model. For base model, training to generate videos exceeding 10 seconds while maintaining high resolution and frame rates is extremely challenging. The main limitation lies in the sequence length. Longer-durating generation requires more advanced video compression technique, which remains for future exploration.
>
> [1] Li, Wenhao, et al. "Training-free Long Video Generation with Chain of Diffusion Model Experts." arXiv preprint arXiv:2408.13423 (2024).
>
> [2] Zhao, Canyu, et al. "Moviedreamer: Hierarchical generation for coherent long visual sequence." arXiv preprint arXiv:2407.16655 (2024).

---

> > ### Comment · Area_Chair_41t7 · 2024-11-24
> > **Discussion Period Ending Soon**
> >
> > Dear Reviewer,
> >
> > The discussion period will end soon. Please take a look at the author's comments and begin a discussion.
> >
> > Thanks, Your AC

---

> ### Author Response · Authors · 2024-11-25
>
> Dear Reviewer,
>
> If our answers above solve your concerns, could you increase your rating?
>
> Thanks, Authors

---

> > ### Comment · Area_Chair_41t7 · 2024-12-01
> > **Discussion Period**
> >
> > Dear Reviewer,
> >
> > Discussion is an important part of the review process. Please discuss the paper with the authors.
> >
> > Thanks, Your AC

---

### Meta-Review · Area_Chair_41t7 · 2024-12-20

**Metareview:**

The paper addresses the problem of video generation and proposes a new model based on a 3D VAE, an expert transformer with expert adaptive LayerNorm, and a progressive training and multi-resolution frame pack technique. The paper also two dataset methods: a video filtering method and a captioning scheme which involves aggregating image captions and using an LLM to meld them into a video caption. The paper demonstrates strong results.

While many of the choices are fairly straight-forward, the analysis on the 2D+1D vs 3D attention as well as the layer norm are useful for the field as well as an open-sourced strong video model. I think the main weakness of the paper is conceptual / technical novelty (in the sense that many of these ideas have either been explored or are very similar to those explored), but given the current interest in video generation and the need for strong open-source video models as well as clear presentation, good results, and useful analysis, I advocate for acceptance.

I recommend that the authors integrate the discussion of the computational cost of 3D vs 2D+1D as well as any of the additional details mentioned.

**Additional Comments On Reviewer Discussion:**

The reviewers generally praised the clarify of the paper as well as the performance of the final model. The weaknesses mostly centered around the lack of certain details (which were addressed in the rebuttal) and as well as the core novelty (xLCm). In general, I agree with the reviewers who highlight the strong performance and find that the weaknesses were relatively minor in comparison.

---

### Decision · Program_Chairs · 2025-01-22

Accept (Poster)